# *Gaze-VLM:* Bridging Gaze and VLMs via Attention Regularization for Egocentric Understanding

**Anupam Pani**[1]    **Yanchao Yang**[1,2]

[1]HKU Musketeers Foundation Institute of Data Science, The University of Hong Kong
[2]Department of Electrical and Electronic Engineering, The University of Hong Kong
apani3@connect.hku.hk, yanchaoy@hku.hk

## Abstract

Eye gaze offers valuable cues about attention, short-term intent, and future actions, making it a powerful signal for modeling egocentric behavior. In this work, we propose a gaze-regularized framework that enhances VLMs for two key egocentric understanding tasks: fine-grained future event prediction and current activity understanding. Unlike prior approaches that rely solely on visual inputs or use gaze as an auxiliary input signal , our method uses gaze only during training. We introduce a gaze-regularized attention mechanism that aligns model focus with human visual gaze. This design is flexible and modular, allowing it to generalize across multiple VLM architectures that utilize attention. Experimental results show that our approach improves semantic prediction scores by up to 11% for future event prediction and around 7% for current activity understanding, compared to the corresponding baseline models trained without gaze regularization. These results highlight the value of gaze-guided training in improving the accuracy and robustness of egocentric VLMs. Overall, this work establishes a foundation for using human gaze to enhance the predictive capabilities of VLMs in real-world scenarios like assistive robots and human-machine collaboration. Code and additional information is available at: https://github.com/anupampani/Gaze-VLM

## 1   Introduction

Vision-Language Models (VLMs) are foundation models that jointly process visual and textual inputs to understand and generate multi-modal information. Foundational works such as ViLBERT, LXMERT, and CLIP have demonstrated their versatility across tasks like image captioning, visual question answering, and multi-modal retrieval (Lu et al., 2019a; Tan and Bansal, 2019; Radford et al., 2021). Beyond these benchmarks, VLMs hold great promise for real-world applications requiring human-machine collaboration, such as assistive robotics (Li et al., 2024), accessibility tools (Zhao et al., 2024b), and autonomous driving (Zhou et al., 2024a). In such domains, temporally grounded tasks, like future action prediction and activity understanding, are critical. Accurate short-term predictions enable systems to anticipate user needs and provide timely, context-aware assistance.

While coarse activity predictions (e.g., "brewing coffee") provide general context, fine-grained descriptions (e.g., "reaching for the coffee capsule in the top-right cabinet") offer actionable detail (Goyal and Durrett, 2021). Achieving this level of understanding requires models to reason about short-term goals as well as visual focus. We posit that *eye gaze* is a powerful supervisory signal in this setting: it reflects a person's attention, often precedes interaction, and encodes both spatial and temporal cues (Frischen et al., 2007; Tipper, 2010).

To leverage these insights, we introduce a gaze-regularized framework that uses gaze only during training to guide attention, enabling deployment with standard image frames. Our method incorporates a gaze-regularized attention block after the visual encoder, aligning attention maps with gaze

distributions from spatial heatmaps using a KL divergence loss. The framework is modular and compatible with various VLMs which use transformer-based architectures.

We validate our approach using several open-source VLM backbones on an egocentric dataset annotated with gaze and fine-grained captions. Our model improves future action prediction by 11% and activity understanding by 7%, compared to baselines trained without gaze. To summarize, **our contributions are:** 1) A modular gaze-regularized VLM framework that improves egocentric activity understanding and generalizes across multiple architectures; 2) A gaze-guided attention mechanism that aligns model attention with human gaze during training; 3) An occlusion-aware filtering strategy that improves the reliability of aggregated spatial heatmaps; 4) A comprehensive evaluation across tasks and models, demonstrating strong performance even without gaze at inference.

## 2 Related Work

Attention mechanisms have become fundamental in vision tasks for identifying salient features, with gaze information serving as a valuable cue in egocentric settings. Prior research has leveraged gaze for tasks such as next-active object prediction (Thakur et al., 2023) and task-relevant information extraction (Hayhoe et al., 2003). Recent transformer-based approaches, including global-local transformers for gaze prediction (Lai et al., 2024) and joint gaze-action modeling (Huang et al., 2020), have demonstrated the effectiveness of capturing gaze dynamics.

The idea of leveraging human gaze to guide computational models is well established. Min and Corso (2020) integrated gaze into an attention mechanism for egocentric activity recognition. They modeled gaze distributions using a variational autoencoder and used it to modulate features, demonstrating that gaze is a powerful signal even when not available at test time. A key distinction lies in the use of optical flow: their two-stream architecture requires flow as a model input during both training and inference, whereas we use flow solely during pre-processing for occlusion-aware gaze filtering, making it unnecessary at test time. While sharing the high-level goal of using gaze as a training-time signal, our work focuses on large-scale VLMs for generating fine-grained textual descriptions. Furthermore, our method presents a simple, modular attention regularization loss that can be plugged into various transformer-based VLMs.

There has been growing interest in aligning VLM behavior with human attention. For instance, Voila-A (Yan et al., 2023) aligns model cross-attention with user gaze during training and inference to steer predictions for static image tasks, requiring gaze input at test time. In contrast, our method uses gaze only during training, making it applicable to standard VLM inference. Other works have explored generating attention maps from human-object interactions (Zhou et al., 2024b) or, in the context of static image classification, using teacher models within contrastive learning frameworks to produce spatial-attention labels for improved robustness (Yao et al., 2023). Our work is situated in the dynamic, temporally-grounded setting of egocentric videos, where we introduce a pipeline for aggregating and filtering raw gaze signals to create robust supervisory heatmaps for regularizing VLM attention during training.

We extend this research by introducing a gaze-regularized attention mechanism that integrates human gaze into VLMs. Rather than predicting gaze, our method uses it during training to shape model attention, aligning computational focus with human viewing behavior. This approach embeds biologically inspired visual priors directly into the attention structure of VLMs, extending gaze applications beyond recognition to multi-modal reasoning and prediction.

Activity understanding and prediction are central to egocentric behavior modeling, with applications in assistive robotics and wearable AI. Recent advances have evolved from LSTM-based methods (Furnari and Farinella, 2019) to hierarchical, intention-conditioned designs (Zhao et al., 2024a; Mascaro et al., 2024) and transformer-based models that capture fine-grained spatiotemporal dependencies (Roy et al., 2024). Building on these foundations, our work incorporates gaze-based attention regularization to enhance context-aware prediction of both current and future activities in egocentric video.

## 3 Method

In this section, we present our approach for enhanced training of VLMs by incorporating human eye gaze data for egocentric activity understanding and future action prediction.

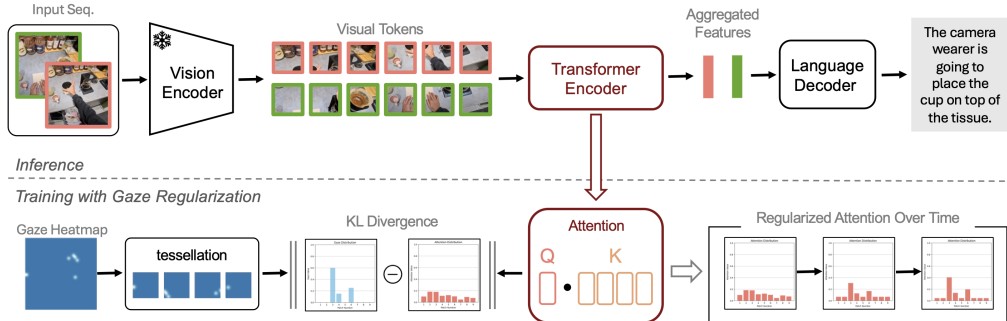

Figure 1: **Overview of the Proposed Scheme.** During inference (**top**), egocentric frames are processed by a vision encoder to extract visual tokens, which flow through a transformer encoder to produce aggregated features that are decoded into descriptive text. During training (**bottom**), we add gaze regularization: human gaze heatmaps are converted to patch-wise distributions matching the transformer's attention granularity. The model's attention distribution is then guided toward human visual focus patterns through KL-divergence minimization, enhancing feature aggregation without requiring gaze data during inference.

Our method builds upon the well-established observation that human gaze patterns reveal crucial information about attention allocation and intention. Rather than treating gaze as merely an additional input feature, we employ it as a principled regularization signal during training to guide the transformer's attention mechanisms. This approach effectively aligns the model's computational attention with human visual focus, creating a more biologically plausible processing mechanism that leverages the correlation between gaze patterns and intentional behavior.

Figure 1 provides an overview of the proposed training pipeline. The system processes egocentric video frames through a vision encoder to extract visual features, which are then modulated through our gaze-regularized attention blocks. During training, the attention mechanism is explicitly regularized to align with human gaze patterns using Kullback-Leibler divergence. The resulting attention-modulated features enable the model to generate descriptive text of current activities or predictions of forthcoming actions, while requiring only standard visual input during inference.

### 3.1 Gaze Representation with Temporal Aggregation

We construct gaze supervision signals for the regularizer by first transforming individual gaze points spatially and then aggregating them temporally with occlusion handling. The gaze points in the dataset are originally represented in text form, and we construct the gaze supervision signals from the coordinates by first transforming them into spatial heatmaps.

**Spatial Transformation** For each gaze point $g_t = (g_t^x, g_t^y) \in [1, ..., \mathrm{w}] \times [1, ..., \mathrm{h}]$ at time $t$, we first generate a spatial heatmap $\mathbf{m}_t \in \mathbb{R}^{\mathrm{h} \times \mathrm{w}}$ through a Gaussian smoothing:

$$\mathbf{m}_t = \pi(G_\sigma * \mathbf{1}(g_t)), \tag{1}$$

where $\mathbf{1}(g_t)$ is an indicator function with value 1 at position $g_t$ and 0 elsewhere, $G_\sigma$ is a Gaussian kernel with standard deviation $\sigma$, $*$ denotes convolution, and $\pi$ represents a normalization so the heatmap sums to 1. This process transforms discrete gaze coordinates into 2D probability distributions that reflect the spatial allocation of visual attention.

**Temporal Aggregation with Occlusion Handling** Human visual attention involves fixations lasting approximately 200ms (Rayner, 2009), interspersed with rapid saccadic movements. Since saccades reflect transitional eye movements rather than focused attention, relying on individual gaze points, particularly those captured during saccades, can introduce noise into the supervision signal. To mitigate this, we propose aggregating spatially transformed gaze maps over a temporal window. This temporal integration captures the stable structure of visual attention and yields a more informative and robust supervisory signal for training. This aggregation over a temporal window is done via:

$$\mathrm{H}_t = \pi\{ \sum_{\tau=t-\delta}^{t} \mathrm{o}_\tau \cdot (f_{\tau \to t} \circ \mathbf{m}_\tau) \}, \tag{2}$$

where $\delta$ defines the temporal window (e.g., 200ms) and $f_{\tau \to t}$ is a warping function (e.g., optical flow) computed from RGB frames that represents pixel motion from time $\tau$ to $t$, and is used to transport $\mathbf{m}_\tau$ to the current frame at time $t$. Furthermore, considering that a point in the scene fixated upon may become occluded due to camera movement (e.g., out of frame) or a change in the environment (e.g., dynamic objects), including such occluded points would create misleading supervision signals.

Therefore, we perform an occlusion check using bidirectional optical flow consistency between frames (Hur and Roth, 2017), resulting in $o_\tau$ that indicates the validity of gaze points to facilitate the temporal aggregation. If the occlusion is significant and major, the gaze point related to the image frame is considered invalid and is not used for the temporal aggregation. The above procedure ensures that our supervision signal

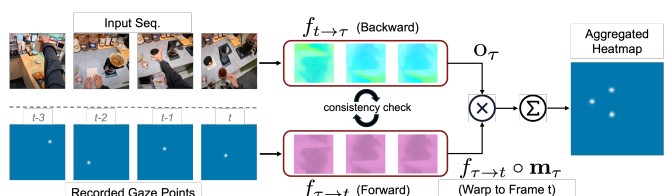

Figure 2: **Temporal Aggregation of Gaze.** *Top:* Input egocentric frames. *Bottom:* Gaze points transformed into heatmaps, filtered using bidirectional optical flow ($f_{\tau \to t}$, $f_{t \to \tau}$) to remove occluded points, then warped and aggregated into a temporally coherent supervision signal.

only incorporates gaze directed at visible elements at the current time step. Please refer to Figure 2 for an illustration of this process. The resulting aggregated heatmaps ($\mathrm{H}_t$) serve as robust supervision targets for the proposed attention regularization framework (more details are provided in Section A.3).

## 3.2 Problem Formulation

We address egocentric behavior understanding through a vision-language approach that enables both interpreting current activities and anticipating future ones. These capabilities are essential for applications like assistive technologies and human-robot collaboration, where machines must comprehend ongoing human actions and predict subsequent behavior.

Given a sequence of egocentric video frames $\{I_t\}_{t=1}^{\tau_o}$ spanning an observation window of $\tau_o$ seconds, our model generates textual descriptions that capture human activities. This framework supports two complementary tasks:

- **Activity Understanding**: Generating descriptions of activities occurring within the observation window of $\tau_o$ seconds.

- **Future Activity Prediction**: Generating descriptions of activities likely to occur in the upcoming $\tau_a$ seconds based on the observed video frames.

Formally, our model learns to generate appropriate textual descriptions $\ell$ conditioned on the observed image sequence:

$$\phi(\ell | \{I_t\}_{t=1}^{\tau_o}) = p(\ell | \{I_t\}_{t=1}^{\tau_o}), \tag{3}$$

where $\ell$ represents either activity descriptions or future activity predictions, depending on the task.

A key aspect of our approach is that the model operates on standard visual inputs (RGB frames) during both training and inference. Gaze information is used exclusively during training as a regularization signal to guide the model's attention mechanism, not as an input feature. This design ensures practical deployability in scenarios where eye-tracking data might be unavailable during inference.

## 3.3 Gaze-Regularized Attention Mechanism

Transformer-based VLMs typically process visual inputs through three main stages: feature extraction via a vision encoder, feature refinement through attention mechanisms, and text generation via language decoding. Our approach targets the attention mechanisms specifically, leaving the vision encoder and language decoding components largely unchanged.

Most VLMs extract visual features $\psi_I = \{\psi_1, \psi_2, ..., \psi_N\}$ from input frames using pre-trained encoders (such as ViT), where each $\psi_i \in \mathbb{R}^d$ represents a feature vector (token) corresponding to an image patch. These features are then used to compute the keys and values in the attention module. For the query, we derive a global query across the input sequence, capturing the overall scene context and

activity information, rather than using frame-specific queries that risks being overfitted to the image photometrics. The global query attends over all spatial patches within an input image to compute attention weights. These weights are later compared against human gaze patterns via the gaze-based regularization. By introducing gaze regularization to the attention computation, we guide the model to focus on regions that humans naturally attend to when performing activities or interactions. Figure 1 illustrates the training pipeline of the proposed gaze-regularized VLMs as well as the regularization process, which shows how the model's attention distribution gets more aligned with the human gaze distribution over time. Next, we elaborate on the details.

**Attention Computation** Our approach utilizes standard attention mechanisms found in transformer architectures. In these implementations, attention is computed as:

$$\text{Attention}(Q, K, V) = \text{softmax}\left(\frac{QK^T}{\sqrt{d_k}}\right) V = AV\,, \tag{4}$$

where $Q$, $K$, and $V$ are query, key, and value derived from visual features, $d_k$ is the dimension of the key vectors, and $A$ represents the attention weights. As mentioned previously, we utilize a global query obtained from the input sequence, whereas the keys and values are derived from individual frames. Using the gaze-based regularization, which is introduced next, we aim to align the model's attention weights more closely with human gaze patterns.

**Gaze Regularization** Before aligning the model's attention with human visual focus, we need to transform the pixel-wise gaze heatmap $H_t$ into a patch-wise distribution that matches the granularity of transformer attention. This transformation is necessary because vision transformers operate on image patches rather than individual pixels, and the attention mechanism computes weights with keys from image patches.

We thus divide the domain $\mathbf{\Omega}$ of the (aggregated) heatmap $H_t$ into a grid of $P$ patches that correspond to the same spatial partition used by the vision encoder (e.g., $\cup_{i=1}^{P}\mathbf{p}_i = \mathbf{\Omega}$ and $\mathbf{p}_i \cap \mathbf{p}_j = \emptyset$), and compute the gaze score for each patch as:

$$\tilde{H}_{t,i} = \frac{1}{Z} \sum_{(x,y)\in\mathbf{p}_i} H_t(x, y)\,, \tag{5}$$

where $i \in \{1, 2, ..., P\}$ is the patch index, and $Z = \sum_{x,y} H_t(x, y)$ is a normalization constant ensuring that $\sum_{i=1}^{P} \tilde{H}_{t,i} = 1$.

These patch-wise gaze scores $\tilde{H}_t$ can now be directly compared to the model's attention weights $A_t$, which operate at the same patch level. The key to the proposed simple yet effective approach is adding a regularization term during training that encourages the model's attention weights to align with this human gaze distribution:

$$D_{KL}(A_t\|\tilde{H}_t) = \sum_{i=1}^{P} A_{t,i} \log \frac{A_{t,i}}{\tilde{H}_{t,i}}\,, \tag{6}$$

where $D_{KL}$ is the Kullback-Leibler divergence between the two patch-wise distributions. This regularization guides the model to attend to regions that humans focus on when performing activities, while still allowing it to learn task-relevant attention patterns from the training data.

**Training Objective** The overall training objective combines the standard cross-entropy loss for text generation with our gaze-based attention regularization term:

$$\mathcal{L}_{\text{total}} = \mathcal{L}_{\text{CE}} + \lambda \cdot \sum_t D_{KL}(A_t\|\tilde{H}_t)\,, \tag{7}$$

where $\mathcal{L}_{\text{CE}} = -\sum p(\ell) \log(\phi(\ell|\{I_t\}_{t=1}^{T_o}))$ is the cross-entropy loss between predicted and ground-truth descriptions, and $\lambda$ is a hyperparameter controlling the strength of the regularization.

This formulation ensures that the model learns to generate accurate textual descriptions while developing attention patterns that align with human visual focus. By adjusting the hyperparameter $\lambda$, we can control the influence of human gaze patterns on the model's attention mechanism, allowing us to systematically evaluate the impact of gaze regularization on egocentric behavior understanding tasks, which we discuss in the following section.

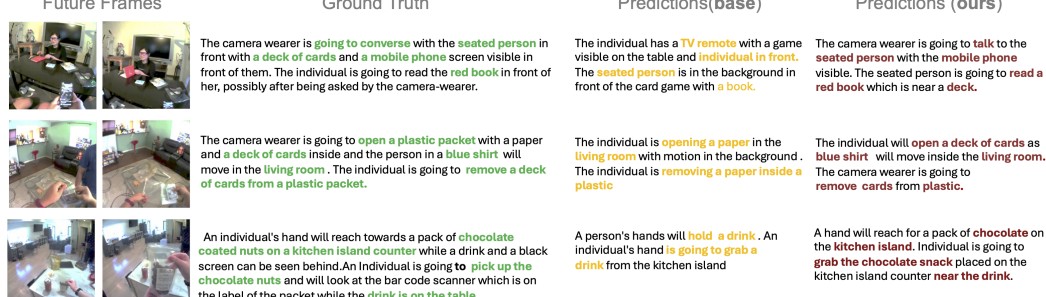

Figure 3: **Qualitative Results of Future Activity Prediction.** Examples comparing predictions from the base model and our gaze-regularized model against ground-truth annotations and actual future frames. The gaze-regularized model correctly predicts specific objects and actions (e.g., "picking up chocolate") while the base model makes less accurate predictions (e.g., incorrectly predicting "a drink"). Key words in the predictions are highlighted to emphasize differences in prediction specificity and accuracy.

## 4 Experiments

We conduct a comprehensive set of experiments to evaluate the effectiveness and generalization of our proposed gaze-regularized framework for egocentric behavior understanding. Our experiments utilize Ego4D clips with gaze annotations, which provide the necessary ground truth for both training and evaluation. The primary experimental comparison contrasts our gaze-regularized models with a base model that uses only RGB inputs and standard attention, without any gaze supervision or alignment during training. To further analyze the effects of gaze regularization, we perform ablation studies by varying its strength, evaluate performance across different temporal anticipation horizons, and examine runtime efficiency for practical deployment. For future prediction tasks, we use an observation window of $\tau_o = 5$s and a prediction horizon of $\tau_a = 2$s, while for current activity understanding, we use a observation window of $\tau_o = 3$s. Our evaluation methodology is based on semantic similarity scores computed using a Sentence-BERT-based transformer (Reimers and Gurevych, 2019), which rewards semantically coherent predictions and penalizes irrelevant or incoherent outputs. In addition to the experiments provided in the main paper, we also provide extensive ablation which can be found in Section A.5 of the Appendix.

### 4.1 Dataset Construction

We adapt the Ego4D dataset (Grauman et al., 2022) to construct a training set suitable for egocentric activity understanding and future prediction using gaze-regularized VLMs. Ego4D provides egocentric video clips with synchronized eye-tracking data, which we leverage through the following processing steps:

1. **Temporal Sampling**: To reduce computational load while preserving temporal structure, we downsample all videos to one frame per second.

2. **Spatial Heatmap Formation**: Raw gaze coordinate annotations (in pixel format) are converted into spatial heatmaps through Gaussian filtering, which transforms sparse gaze points into continuous attention distributions that better reflect the spread and uncertainty of human visual focus. These heatmaps serve as soft supervisory signals during training, allowing the gaze regularizer to guide the model's attention toward regions that align with human perceptual patterns.

3. **Textual Description Generation**: We use GPT-4V (OpenAI, 2023) to generate fine-grained textual descriptions that capture the egocentric activities. To ensure temporal consistency and contextual relevance, we provide GPT-4V with the image sequences and iteratively refine the prompts. The final prompt template, obtained after multiple rounds of feedback and validation, guides GPT-4V to generate descriptions that include objects being manipulated, ongoing actions, and spatial movement cues. This process results in the formation of finer-grained annotations which we utilize to train the model for the egocentric behavior understanding task.

Together, these steps yield a training dataset consisting of synchronized RGB frames, gaze heatmaps, and rich textual annotations. This curated dataset provides the necessary visual, attentional, and

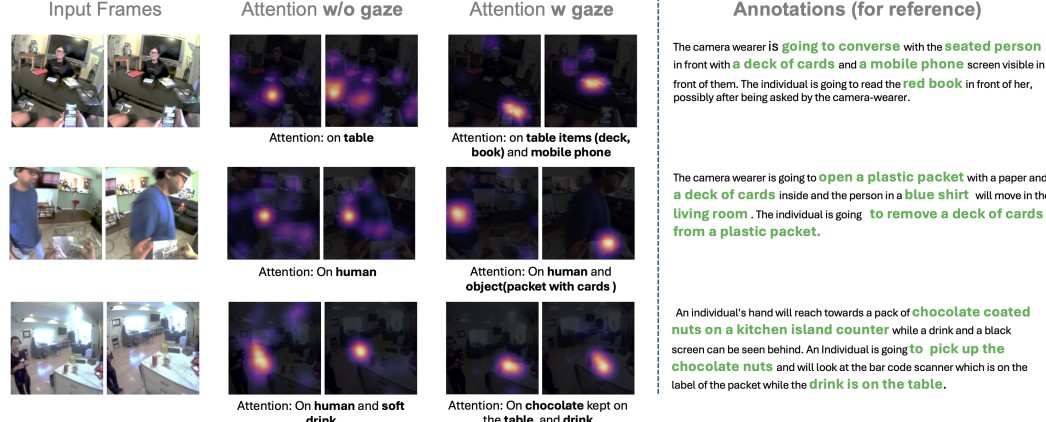

Figure 4: **Comparison of Attention Maps.** Attention distributions from base model (without gaze regularization) versus our gaze-regularized model. Our approach produces more semantically meaningful attention: *Top:* Our model focuses on specific objects (cards, book, phone) while the base model attends broadly to the table. *Middle:* Our model attends to both the person and plastic packet, compared to the base model's person-only focus. *Bottom:* Our model correctly attends to chocolate and drink on the counter, while the base model attends to less relevant areas. This demonstrates how gaze regularization guides the model to focus on objects crucial for understanding human activities.

semantic supervision to effectively train our gaze-regularized vision-language models for both current activity understanding and future event prediction in egocentric settings. More information about the dataset can be found in Section A.2 of the Appendix.

## 4.2 Gaze-Regularized Model Evaluation Across Architectures

We evaluate the effectiveness of our gaze-regularized framework across five transformer-based VLM architectures: OpenFlamingo, a modified OpenFlamingo without the Perceiver Resampler, LaViLa's Narrator module (Zhao et al., 2022), InternVL (2.5-1B) (Chen et al., 2024), and OpenLLaVA (LLaVA-NeXT-Vicuna-7B)(Lin and Long, 2024). Each model is trained on the same Ego4D-based dataset and evaluated using RGB-only inputs for two tasks: future event prediction and current activity understanding. Our framework integrates a gaze-regularized attention block between the visual encoder and the language decoder, applying gaze supervision exclusively during training. This modular design makes it applicable to any architecture that utilizes attention mechanisms, while maintaining practical inference by eliminating any dependency on gaze data at test time.

As reported in Table 1, our method consistently improves performance across all architectures. For future prediction, we observe absolute gains ranging from **8.9% to 10.5%**, with OpenFlamingo and its modified variant showing the strongest improvements. In current activity understanding, we see improvements between **4.9% and 6.9%**, with LaViLa achieving the largest increase. These results demonstrate that gaze-guided regularization meaningfully enhances temporal and contextual reasoning in egocentric vision tasks. Figure 3 shows the annotations predicted by the models, while Figure 9 visualizes attention maps for qualitative comparison.

## 4.3 Sensitivity to Gaze Regularization Scale

To assess the impact of the gaze-regularization term, we conduct an ablation study by systematically varying the regularization strength parameter $\lambda$ during training. When $\lambda=0$, the model is trained without any gaze supervision, using only the cross-entropy loss, though the rest of the architecture remains unchanged.

As shown in Table 2, a moderate regularization strength ($\lambda=100$) consistently yields the best results across both tasks and all architectures. The most significant performance drop occurs in the unregularized setting ($\lambda=0$), underscoring the value of gaze supervision in improving attention allocation and prediction quality. While increasing $\lambda$ to 1000 still leads to competitive results, it introduces a marginal performance decline in some cases as compared to the setting ($\lambda=100$).

Table 1: Performance comparison of baseline models versus our gaze-regularized approach across five VLM architectures. Results show semantic similarity scores for future prediction and activity understanding tasks on Ego4D. All models use RGB-only inputs at inference time, demonstrating consistent performance gains (4.9-10.5%) with our method across all architectures and tasks.

| MODEL | FUTURE PREDICTION | | | ACTIVITY UNDERSTANDING | | |
|---|---|---|---|---|---|---|
| | BASE | OURS | GAIN | BASE | OURS | GAIN |
| OPENFLAMINGO | 0.6525 | **0.7505** | +9.8% | 0.7176 | **0.7848** | +6.7% |
| MODIFIED OPENFLAMINGO | 0.6155 | **0.7200** | +10.5% | 0.6677 | **0.7300** | +6.2% |
| LAVILA NARRATOR | 0.6030 | **0.6924** | +8.9% | 0.6576 | **0.7268** | +6.9% |
| INTERNVL | 0.6323 | **0.7318** | +9.9% | 0.6831 | **0.7404** | +5.7% |
| OPENLLAVA | 0.5872 | **0.6771** | +9.0% | 0.6539 | **0.7027** | +4.9% |

Table 2: Effect of regularization strength ($\lambda$) on semantic performance across VLM architectures.

| MODEL | FUTURE PREDICTION | | | ACTIVITY UNDERSTANDING | | |
|---|---|---|---|---|---|---|
| | $\lambda=0$ | $\lambda=100$ | $\lambda=1000$ | $\lambda=0$ | $\lambda=100$ | $\lambda=1000$ |
| OPENFLAMINGO | 0.6317 | **0.7505** | 0.7354 | 0.6917 | **0.7848** | 0.7721 |
| MODIFIED OPENFLAMINGO | 0.6032 | **0.7200** | 0.7072 | 0.6603 | **0.7300** | 0.7194 |
| LAVILA NARRATOR | 0.6033 | **0.6924** | 0.6765 | 0.6310 | **0.7268** | 0.7123 |
| INTERNVL | 0.6342 | **0.7318** | 0.7190 | 0.6867 | **0.7404** | 0.7287 |
| OPENLLAVA | 0.5882 | **0.6771** | 0.6628 | 0.6523 | **0.7027** | 0.6895 |

These findings suggest that while gaze-based alignment substantially benefits model performance, excessive regularization may constrain the model's ability to attend to regions that are task-relevant but not explicitly captured by human gaze. We systematically explored different scaling values to determine the range in which the regularizer is most effective, though further investigation may be needed to precisely calibrate optimal regularization strength across different tasks and architectures.

## 4.4 Impact of Anticipation Window Length on Predictive Accuracy

We further evaluate the robustness and temporal generalization of our approach by varying the anticipation window $\tau_a$, testing the model's ability to predict future activities at different time horizons. Specifically, we compare performance at 2-second and 5-second intervals, using a fixed observation window $\tau_o = 5s$ across both settings. This experiment allows us to assess whether the benefits of gaze regularization persist as the prediction task extends further into the future, where the increasing temporal gap between observation and target introduces greater uncertainty.

Table 3: Performance across anticipation horizons for future event prediction with two different variations.

| MODEL | $\tau_a = 2s$ | | $\tau_a = 5s$ | |
|---|---|---|---|---|
| | BASE | OURS | BASE | OURS |
| OPENFLAMINGO | 0.6525 | **0.7505** | 0.6297 | **0.7315** |
| MOD. OPENFLAMINGO | 0.6155 | **0.7200** | 0.5911 | **0.7094** |
| LAVILA NARRATOR | 0.6030 | **0.6924** | 0.5888 | **0.6713** |
| INTERNVL | 0.6323 | **0.7318** | 0.6172 | **0.7004** |
| OPENLLAVA | 0.5872 | **0.6771** | 0.5491 | **0.6492** |

As shown in Table 3, our gaze-regularized model consistently outperforms the baseline at both horizons. The performance advantage is particularly notable because human gaze patterns can provide early indicators of intent that help bridge the temporal gap between current observations and future actions. Nevertheless, we observe that performance for both models declines when the prediction horizon increases from 2 to 5 seconds. This decline is expected due to the inherent challenges of longer-horizon prediction, where the available context from the observation window becomes less informative and the range of possible future actions expands, making accurate prediction increasingly difficult.

Table 4: Out-of-distribution generalization on EGTEA+ Gaze. All models are trained on our dataset and evaluated using RGB-only inputs during inference.

| MODEL | FUTURE PREDICTION | | | ACTIVITY UNDERSTANDING | | |
| --- | --- | --- | --- | --- | --- | --- |
| | BASE | OURS | GAIN | BASE | OURS | GAIN |
| OPENFLAMINGO | 0.6501 | **0.7305** | +8.0% | 0.6963 | **0.7527** | +5.6% |
| MODIFIED OPENFLAMINGO | 0.6071 | **0.6797** | +7.3% | 0.6326 | **0.6924** | +5.0% |
| LAVILA NARRATOR | 0.5804 | **0.6632** | +8.3% | 0.6397 | **0.7003** | +6.1% |
| INTERNVL | 0.6230 | **0.7107** | +8.8% | 0.6712 | **0.7265** | +5.5% |
| OPENLLAVA | 0.5714 | **0.6209** | +4.9% | 0.6338 | **0.6615** | +2.8% |

## 4.5 Evaluating Generalization on Out-of-Distribution Egocentric Data

To assess how well our gaze-regularized framework generalizes beyond the training distribution, we evaluate it on the EGTEA+ Gaze dataset (Li et al., 2020a). This setting allows us to test whether the attention alignment learned from Ego4D transfers effectively to new egocentric domains.

We evaluate both the base and gaze-regularized models on this new dataset without any additional fine-tuning, using only models trained on our original Ego4D-derived dataset. As shown in Table 4, the gaze-regularized models maintain their performance advantages, achieving improvements of **7–9%** in future prediction and **5–6%** in activity understanding across most architectures. While we observe some performance variations across models, these can partly be attributed to the adaptations required to make each architecture compatible with our input pipeline. These results suggest that our method produces attention patterns that generalize beyond the training domain and remain effective in novel egocentric settings. We note that these comparisons serve primarily as a sanity check rather than a definitive ranking of model capabilities across datasets.

## 4.6 Runtime–Performance Analysis of Gaze Regularization

Inference efficiency is essential in real-world applications such as assistive robotics, where latency and computational constraints must be carefully balanced. To evaluate this critical performance-efficiency trade-off, we measure both the semantic accuracy and runtime characteris-

Table 5: Runtime performance of various model variants from the openflamingo architecture.

| MODEL VARIANT | SCORE | GAIN | RUNTIME (S) |
| --- | --- | --- | --- |
| BASE (NO REG.) | 0.6525 | – | 1.7 |
| GAZE-REG (W/O OCCL.) | 0.7298 | +7.7% | 2.3 |
| GAZE-REG (W/ OCCL.) | **0.7505** | +9.8% | 2.3 |

tics of different OpenFlamingo variants under our gaze-regularization framework, as OpenFlamingo consistently demonstrated the strongest overall performance in our experiments.

During training, the inclusion of occlusion filtering adds computational overhead, as it requires identifying and discarding unreliable gaze inputs to improve the quality of supervision. However, at inference time, all models operate using only RGB inputs–no gaze data, heatmaps, or occlusion computation is required. This design choice ensures that our approach maintains practical deployment efficiency while benefiting from gaze-guided learning during the training phase.

As shown in Table 5, our best-performing model (Gaze-Reg w/ occlusion) achieves a **9.8%** gain in semantic performance over the base model, with only a minor increase in inference time (2.3s vs 1.7s). These measurements reflect forward pass time only and exclude pre-processing or data loading steps. All tests conducted using an NVIDIA A800 GPU to ensure consistent evaluation conditions.

## 4.7 Reducing Visual Hallucinations

We further investigate whether gaze regularization mitigates visual hallucinations - a known issue where VLMs generate descriptions containing objects or actions absent from the visual input. Standard benchmarks for evaluating hallucinations, such as HallusionBench (Guan et al., 2024), provide established metrics for this phenomenon but lack the gaze annotations required for our method.

Since standard hallucination benchmarks lack gaze annotations, we conducted a controlled human evaluation. We selected 200 examples from our dataset and generated outputs from both the baseline and gaze-regularized models. These responses were randomly shuffled and presented to evaluators blind to the model source, who were shown the ground-truth video context and instructed to flag any instances of clearly hallucinated or unrelated content. Taking inspiration from the metric in HallusionBench, we report the $C_I$ score, which is the ratio of model responses containing clearly hallucinated instances to the total number of responses evaluated.The results, summarized in Table 6, show that our method meaningfully reduces hallucination. The $C_I$ score dropped from **0.205** to **0.140**. This supports the hypothesis that by focusing the model's attention on visually-grounded, human-attended regions, our approach discourages the model from "inventing" details based on linguistic priors alone, leading to more reliable and trustworthy responses.

Table 6: Human evaluation of visual hallucination rates. Our gaze-regularized model reduces the occurrence of hallucinated content, as measured by the $C_I$ score.

| Model Variant | Hallucinated Cases | Total Samples | $C_I$ |
|---|---|---|---|
| Base Model | 41 | 200 | 0.205 |
| Gaze-Regularized Model | **28** | 200 | **0.140** |

### 4.8 Quantifying Attention-Gaze Alignment

A core premise of our work is that aligning a model's attention with human gaze leads to better task performance. To quantitatively validate that our gaze-regularized model indeed learns to approximate human attention patterns, we measured the spatial overlap between the model's final-layer attention maps and the ground-truth human gaze heatmaps on the test set. Specifically, we computed the top-10 overlap, which measures the proportion of the model's top-10 attended image patches that fall within the human gaze distribution.

The results confirm an improvement in alignment when comparing the base model and the gaze-regularized model. The gaze-regularized model achieved a 68% top-10 overlap, which is an increase from the 42% overlap observed in the base model which does not use gaze regularization. This improvement provides concrete evidence that our regularization loss successfully steers the model's focus toward regions that humans find important. This learned alignment with human priors is a key factor behind the consistent performance improvements observed across all tasks and architectures.

## 5 Conclusion

In this work, we have demonstrated that incorporating human gaze data as a training-time supervisory signal significantly enhances the performance of VLMs in egocentric behavior understanding, improving both current activity recognition and future event prediction. By aligning model attention with human visual focus through gaze-based regularization, our approach captures subtle, temporally grounded cues that static attention mechanisms often miss. Crucially, it requires no gaze input at inference time, making it practical for potential real-world deployment in assistive systems, wearable AI, and human-robot collaboration. The framework is modular and integrates seamlessly with attention-based VLMs such as OpenFlamingo, LaViLa, InternVL, and OpenLLaVA. Across all models and tasks, we observe improvements – up to 11 % in future prediction and 8 % in activity understanding – demonstrating the effectiveness and generalization of our method. Furthermore, this attention alignment substantially improves output reliability, reducing the rate of visual hallucinations based on human evaluation. This suggests that human gaze provides a spatial prior that effectively grounds language generation, reducing reliance on potentially misleading textual associations. Broader adoption of gaze-augmented learning depends on access to large-scale, high-quality datasets with reliable and calibrated gaze annotations. Training on such data with task-specific labels could further enhance performance. Future work may explore joint modeling of gaze and action, as well as extending the framework to other temporally grounded tasks beyond egocentric understanding. We will release our code and dataset to support continued research and encourage further exploration of gaze as a powerful supervisory signal for aligning vision-based models,including VLMs – more closely with human attention and intent.

## Acknowledgments and Disclosure of Funding

This work is supported by the Early Career Scheme of the Research Grants Council (RGC) grant # 27207224, the HKU-100 Award, a donation from the Musketeers Foundation, and in part by the JC STEM Lab of Robotics for Soft Materials funded by The Hong Kong Jockey Club Charities Trust. We would like to extend our sincere gratitude to Siyan Dong, Qihang Fang, Ruizhe Liu and Qian Luo (from HKU IDS) for their invaluable technical discussions and implementation support throughout this project. We also like to thank Andrew Lin from UC Irvine for helping with annotation work and feedback during the experimental phase. Finally, we appreciate the constructive feedback from our reviewers, which helped strengthen this paper and will pave the way for future projects.

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

# A   Appendix

This appendix presents supplementary material to support our study. We provide a table of notations to aid the readers, an expanded discussions of related work, further details on the dataset curation and experimental prompts, and results from supplementary ablation studies. We conclude with comprehensive model training details to facilitate replication of our work.

**Short table of Notations**   To ensure that the readers can directly refer to a table for certain notations without scouring through the text, we provide a table of notations for quick reference where the symbols are accompanied by a short description.

Table 7: Summary of key notations used in temporal gaze aggregation and occlusion filtering.

| Symbol | Description |
|---|---|
| $t$ | Timestamp of current frame |
| $\delta$ | Temporal aggregation window size (e.g., 200ms) |
| $I_t$ | RGB image frame at time $t$ |
| $g_t$ | Gaze point (pixel coordinate) at time $t$ |
| $m_t$ | Spatial heatmap generated from $g_t$ via Gaussian smoothing |
| $H_t$ | Aggregated gaze heatmap at time $t$ after occlusion filtering |
| $f_{\tau \to t}$ | Forward optical flow from frame $I_\tau$ to $I_t$ |
| $f_{t \to \tau}$ | Backward optical flow from $I_t$ to $I_\tau$ |
| $\mathbf{p}$ | Designated pixel location in image $I_\tau$ |
| $\hat{\mathbf{p}}$ | Translated pixel location in $I_t$ using forward flow |
| $\Delta$ | Discrepancy between forward and backward flow vectors |
| $\eta_{\text{observed}}$ | Proportion of pixels with flow discrepancy $> \epsilon$ |
| $\epsilon$ | Threshold for pixel-level flow discrepancy |
| $\eta$ | Threshold for deciding major occlusion (set to 60%) |
| $o_\tau$ | Binary occlusion validity flag for frame $I_\tau$ |

## A.1   More Related Work

**Vision Language Models** VLMs take in as input images and text together $(image, text)$ and produce a text output. VLMs learn a mapping function $(image, text) \to text$ where the task varies from Visual Question Answering (VQA) tasks to generating text based on image and text provided. The VLMs output text condition on the image text sequence and several models exists such as BLIP, LLaVa , Flamingo etc. Liu et al. (2023), Li et al. (2022),Zhang et al. (2023),Alayrac et al. (2022),Chen et al. (2023). In this study, we initially employ the open-source version of the Flamingo model Awadalla et al. (2023), built upon the foundations of the original Flamingo described in Alayrac et al. (2022). To show that our approach can be implemented and generalized to other architectures, we also utilise LaViLa's Narrator module (Zhao et al., 2022), and adapted versions of InternVL(Chen et al., 2024) and OpenLLaVA (Lin and Long, 2024) in our experiments. These models were chosen for their use of attention mechanisms, which are central to our method–our framework explicitly aims to modulate attention to better align with human visual focus. In addition to traditional input modalities, such as RGB images, recent advancements have highlighted the potential of integrating various modalities beyond vision, including gaze, gait, and tactile sensors (Boshoff et al., 2024; Yang et al., 2024). Building on this premise, our approach incorporates eye gaze as an additional signal.

**Dataset** The Ego4D and EPIC-Kitchens datasets consist of egocentric videos of camera-wearers performing daily activities in semi-controlled environments (Grauman et al., 2022; Damen et al., 2022). Additional relevant datasets include the EGTEA+ Gaze dataset, with 28 hours of cooking-centric content paired with gaze (Li et al., 2020b), and the Visual Data Experience (VDE) dataset, which provides approximately 240 hours of everyday activity recordings with synchronized gaze and head tracking (Greene et al., 2024).

While some datasets provide coarse labels, the gaze-augmented subset of Ego4D used in our work lacks fine-grained annotations. To address this, we supplement it with descriptive captions generated via GPT-4V. Future work may extend this setup by incorporating VDE or other multimodal datasets.

Additionally, Huang et al. (2025) introduce a dataset combining egocentric and exocentric perspectives for procedural activities. This dual view setup which aligns with an "observe first, imitate next" paradigm may enhance downstream modeling but the inclusion of this study is left for future work.

**Attention and gaze-augmented models**: Attention-based models are widely used to identify important features and improve performance across various domains, including autonomous driving (Braunagel et al., 2017), action prediction, and human-computer interaction (Weber et al., 2020; Shafti et al., 2019; Aronson et al., 2018). These models enhance interpretability by directing focus to the most relevant regions of an image or sequence.

For instance, class activation maps have been employed to leverage pooling layers in deep networks, generating class saliency maps (Sudhakaran and Lanz, 2018) that highlight key objects associated with future actions. Guided-attention mechanisms have further been used to model next-active object interactions, improving action anticipation (Thakur et al., 2023). The Spatiotemporal Attention Module (STAM) (Lu et al., 2019b) incorporates eye gaze as supervision, training a network to predict attention maps for activity recognition. Other studies have similarly used gaze to guide attention maps in activity recognition tasks (Min and Corso, 2020; Awale and Sarikaya, 2022).

Beyond artificial systems, human perception and action have long been studied in relation to gaze. Eye movements provide task-specific visual cues, enabling more efficient action execution (Hayhoe et al., 2003). In our work, we explore how eye gaze data can be integrated into Vision-Language Models (VLMs) to improve egocentric behavior understanding, conditioning text annotations on gaze data to enhance future activity prediction.

Recent works such as OPERA (Huang et al., 2024) and Perception in Reflection (Wei et al., 2025) operate on the cross-attention maps during text generation and apply retrospective corrections if and when the model focuses on irrelevant regions. For instance, OPERA penalizes captions that mention objects like "pencil" if the cross-attention never highlighted any pencil in the image. Perception in Reflection similarly evaluates whether the model has missed out on finer details by reflecting on generated answers using a multi-turn dialogue dataset.

In contrast, our work focuses on egocentric video understanding, where the goal is to model human behavior and intent in dynamic scenes. In this context, human gaze offers a strong and temporally grounded prior for what the person is doing or about to do as humans naturally fixate on task-relevant objects during interaction. Rather than relying on post-hoc attention correction, our method introduces gaze alignment to shape the visual representation passed into the language module, helping the model focus on semantically meaningful content from the start.

To summarize, inn our project, we study various ways to utilize eye gaze data in a VLM setting, building a gaze-regularized attention mechanism. Our model conditions predictions on eye gaze data as an input signal to output fine-grained text annotations of events happening in the near future, as well as providing detailed annotations of the current activity.Our approach is compatible with various transformer-based VLMs and does not require gaze at inference time.

## A.2 Dataset Creation

This section provides details on the dataset creation process for model training and testing, including the utilization of an occlusion filtering mechanism to create aggregated spatial heatmaps.

### A.2.1 Dataset Construction and Prompt Design

The Ego4D dataset comprises of egocentric video clips along with supplementary data such as audio, text annotations, eye gaze data, and additional metadata (Grauman et al., 2022). For our project, we focused on the subset of video clips that include eye gaze data, containing approximately 33.3 hours of egocentric videos recorded from 80 participants. The eye gaze data is provided in numerical form, containing canonical timestamps and the pixel coordinates of gaze points. We transform gaze points to images to represent important visual regions, aligning with how humans perceive spatial information (Laeng et al., 2014). Due to the minute differences between consecutive images in the original videos, we perform downsampling to one image per second to reduce computational requirements while remaining effective. Since our focus is fine-grained egocentric behavior understanding, we modify existing data with detailed textual descriptions to enhance human-machine interaction. We leverage GPT-4V to generate annotations for video frames by processing a sequence of images and

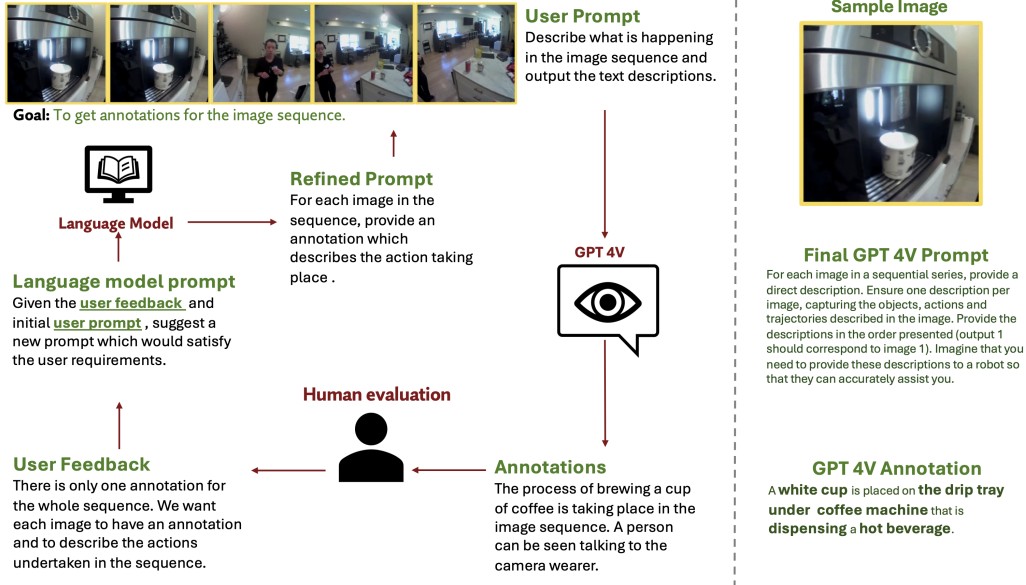

Figure 5: **GPT-4V Annotation Workflow** *Left*. We illustrate our annotation pipeline, where sequences of egocentric images are provided to GPT-4V along with an iteratively refined prompt. Human feedback is used to improve the prompt over multiple rounds, ensuring greater accuracy and contextual coherence in the generated captions. The use of the entire sequence helps maintain temporal consistency, allowing GPT-4V to better capture object interactions and ongoing actions. *Right*. We show a sample image from the dataset, the final refined prompt, and the resulting annotation. This process results in finer-grained text annotations which we utilise in our experiments.

prompting it to describe each frame. This method ensures contextual coherence across frames. After obtaining initial descriptions, we evaluate their quality and provide feedback to refine the output. This feedback is used to modify the prompt, which is then fed back into GPT-4V with the image frames. We conduct prompt selection and refinement using multiple sequences from different video clips to ensure generalization. This iterative process continues until we establish an optimal prompt template that consistently yields accurate and contextually appropriate annotations.

Specifically, we start by passing a small set of images (validation set) to GPT-4V with a basic prompt like: "Describe what is happening in the image sequence and output the text descriptions." The initial output was manually evaluated, and feedback was provided. This feedback, along with the original prompt, was passed to ChatGPT to refine the prompt for generating accurate and meaningful text annotations. The manual evaluation process ensured that the generated annotations met the following criteria:

1. A clear description of the objects being manipulated or focused on in the scene.

2. A detailed account of the actions being performed by both the camera wearer and other individuals present in the images.

3. Information about any trajectories or movements that take place within the scene.

4. Clear, fine-grained annotations that fulfill these criteria in a way that is easily understandable by both humans and machines, such as robots that may use these instructions for task execution.

After several iterations of refining the prompt and evaluating the results, we identified a suitable template for generating high-quality text annotations. This final template was then used to annotate the image sequences using GPT-4V. More details on the prompt design process can be found in Figure 5, and for a corresponding high level view on the algorithm, please refer to Algorithm 1.

We also provide additional details regarding the dataset used in our work. To reiterate, the purpose of using a third-party captioning process was to enhance the granularity of the usual action describing annotations i.e make it more fine-grained. We had approximately 33 hours of data and since the

images are sampled at every 1 second, we had a little over 108000 images.The images were divided into 'chunks' each of about size 10 such that when they are provided to GPT-4V, GPT-4V is aware of the full context. Due to connection issues, sometimes we would get an error and so for such cases, we would discard the chunk. In addition, certain annotations also contained the word 'blurred' or 'unclear', which is not useful for egocentric behavior understanding tasks and we tried to avoid such annotations-image pair as much as possible. Annotations typically ranged from 15–20 words and terms like 'individual' and 'camera-wearer' were common. Their overuse might have contributed to some annotation inaccuracies.

Two human evaluators conducted a comprehensive quality assessment throughout the annotation process. During prompt refinement, they evaluated 500 image-caption pairs across multiple iterations. For the final dataset, we implemented a rigorous quality control protocol: each sequence of frame-wise captions was reviewed, and sequences where more than 20% of frames contained flagged annotations (e.g., object/action mismatches, spurious hallucinations, or overly generic descriptions) were discarded. This resulted in an 80% quality acceptance threshold at the sequence level.

To summarize, the quality assessment occurred in two stages: (1) during prompt refinement, evaluators focused on identifying systematic issues with annotation quality across multiple clips; (2) for the final dataset where generated sequences underwent manual review using explicit flagging criteria. Annotations were flagged if they contained: object/action mismatches, spurious hallucinations, overly generic descriptions, lack of temporal grounding, or terms like 'unclear'/'blurred'. Only sequences with at least 80% unflagged frame-level captions were retained for training, ensuring high-quality supervision signals while maintaining scalability.

**Prompt Refinement Examples**   We arrived at our final, optimized prompt through an iterative process designed to enforce frame-wise specificity and rich scene description. The key stages of this evolution were as follows:

**Initial Prompt:** "Describe what is happening in these images." **Output:** "A person is making coffee in a kitchen and then talking to another individual nearby." **Feedback:** Too generic and summary-like; lacks per-frame grounding and temporal consistency.

**Intermediate Prompt:** "Describe what the person is doing in each frame one by one." **Output:** "Frame 1: No person can be seen in the frame. Frame 2: The person moves toward the counter. Frame 3: The person is talking to another individual." **Feedback:** Overly focused on person visibility rather than comprehensive scene understanding.

**Final Optimized Prompt:** "For each image in the sequential series, provide a direct description. Ensure one description per image, capturing the objects, actions, and their spatial relationships. Provide the descriptions in the order presented (output 1 should correspond to image 1). Imagine that you need to provide these descriptions to a robot assistant." **Output:** "Frame 1: A white cup is placed on the drip tray under the coffee machine, which is brewing and dispensing a hot beverage. Frame 2: The individual has moved closer to the coffee machine and is brewing a cup of coffee. Frame 3: A person in a black shirt can be seen talking to the camera wearer in the kitchen near the coffee machine."

### A.2.2   Annotation-Gaze Alignment

To quantify annotation-gaze alignment, we conducted a separate small scale study where human evaluators judged whether GPT-4V descriptions accurately referenced objects/actions in gaze-attended regions. We provided the evaluators with the original image, as well as a gaze-overlaid image where the gaze occupied regions (heatmap) were overlaid onto the original image, along with the annotation generated. Under strict matching criteria (i.e the gaze regions accurately match the object/actions in the annotations, and are relevant) , 74% of annotations showed precise alignment. The remaining cases included partial alignments (describing relevant context but missing specific gaze-fixated objects) and approximate alignments (capturing general scene context). This conservative estimate demonstrates the overall reliability of our supervision signal, particularly given that our method leverages temporal sequences where gaze-semantic alignment naturally improves across frames.

### A.3   Occlusion-Check Process

The input image sequence can exhibit dynamic changes due to movement in the environment or the movement of the camera wearer. The method for aggregating gaze points is suitable only when the

frames within the $\delta$ interval for which aggregation is done, shows moderate movement. However, preventing movement in a dynamic environment is challenging. If we aggregate gaze points in the $[t - \delta, t]$ interval to construct the aggregated heatmap $H_t$ and there is major occlusion between the earlier frames $[t - \delta, t)$ and the final frame at timestamp $t$, it becomes impractical to include gaze points from the occluded frames. In such cases, collecting gaze points from occluded frames may lead to inaccurate or misleading representations in the heatmap.

To alleviate this issue, we perform an occlusion check between each frame in the $[t - \delta, t)$ interval and the final frame at time $t$ to ensure appropriate gaze aggregation. In the case of significant occlusion or a drastic change in the scene, gaze points corresponding to the earlier frames should not be collected for the aggregation and subsequent formation of the heatmap $H_t$.

Using a method similar to the consistency check with optical flow presented by Hur and Roth (2017), we explicitly exclude gaze points that are occluded in the current frame. If a pixel is correctly translated and there is no major occlusion, then the difference between the forward optical flow displacement of pixel $\mathbf{p}$ and the displacement of the translated pixel $\hat{\mathbf{p}}$ with backward optical flow should be close to zero.

For an RGB image $I_t$ at time $t$, we gather the image frames $\{I_\tau\}$ for all $\tau \in [t - \delta, t)$. Let the forward optical flow between images $I_\tau$ and $I_t$ in the horizontal direction be denoted by $f_{\tau \to t}$ and the backward optical flow by $f_{t \to \tau}$. Let the coordinates of a designated pixel be $p$. The new coordinates of the translated pixel in the subsequent frame, using optical flow, are computed as follows:

$$\hat{\mathbf{p}} = \mathbf{p} + f_{\tau \to t}(\mathbf{p}) \tag{8}$$

Next, we calculate the distance moved by this designated pixel in the horizontal and vertical directions according to the following equations:

$$\Delta = \|f_{\tau \to t}(\mathbf{p}) + f_{t \to \tau}(\hat{\mathbf{p}})\| \tag{9}$$

If the observed proportion of pixels $\eta_{\text{observed}}$ exceeding the distance discrepancy is more than a predefined threshold $\eta$, we conclude that a major occlusion has occurred; otherwise, the occlusion is minor. The observed proportion of such pixels $\eta_{\text{observed}}$ is calculated as:

$$\eta_{\text{observed}} = \frac{1}{H \times W} \sum_{i=1}^{H \times W} \mathbf{1}\left(\Delta_i > \epsilon\right) \tag{10}$$

We disregard the gaze points for frames $\{I_\tau\}$ where there is major occlusion with respect to the image frame $I_t$ and we represent this filtering using the function $o_\tau$ If the occlusion is minor, the appropriate gaze points $\{g_\tau\}$ for all $\tau \in [t - \delta, t]$ are then translated into their new coordinates using the forward optical flow, and collected for the formation of heatmap $H_t$ and subsequently.

As mentioned above, the idea is that if there is a major occlusion, the difference between the distance traversed by a pixel during forward optical flow, and the distance traversed by the translated pixel during backward optical flow will be significantly greater than in cases where the occlusion is minor. Optical flow was calculated using the implementation of the RAFT model developed by Teed and Deng (2020). The hyperparameter $\epsilon$ is the threshold distance, which was set to 20, whereas $\eta$ is the threshold proportion of pixels that have exceeded the occlusion limit, set to 0.60. A brief overview of the process can also be seen in Algorithm 2. While optical flow has been employed in prior work for egocentric vision tasks (Min and Corso, 2020), our approach fundamentally differs in both implementation and computational requirements. Previous methods typically integrate optical flow computation directly into their model architecture, requiring flow estimation during both training and inference phases. In contrast, we leverage optical flow exclusively during the pre-processing stage to enable temporal aggregation of gaze signals across adjacent frames. This allows us to construct more robust gaze heatmaps by compensating for minor head movements and aligning fixations across time. Crucially, by confining optical flow to pre-processing, our method eliminates the need for computationally expensive flow models like RAFT during inference, resulting in significantly faster deployment and reduced computational overhead at test time.

**Algorithm 1:** Iterative Prompt Refinement for Fine-Grained Annotation

---

**Input:** Validation image subset $\mathcal{V} = \{\mathcal{V}_1, \ldots, \mathcal{V}_m\}$;
Full image set $\mathcal{I} = \{\mathcal{I}_1, \ldots, \mathcal{I}_k\}$;
Initial prompt $P_0$;
GPT-4V model; ChatGPT interface
**Output:** Final refined prompt $P^*$;
Annotations $\{\mathcal{T}_1, \ldots, \mathcal{T}_k\}$

**1** $P \leftarrow P_0$;
**2 repeat**
**3**     Generate annotations $\hat{\mathcal{T}}_\mathcal{V} \leftarrow \texttt{GPT4V}(\mathcal{V}, P)$;
**4**     Manually evaluate $\hat{\mathcal{T}}_\mathcal{V}$ for:

- Object relevance and accuracy

- Actions of camera wearer and others

- Movements or spatial cues

- Clarity and fine granularity

    Provide feedback $F$ on $\hat{\mathcal{T}}_\mathcal{V}$;
    Refine prompt: $P \leftarrow \texttt{ChatGPT}(P, F)$;
**5 until** *annotations on $\mathcal{V}$ are satisfactory*;
**6** $P^* \leftarrow P$;          `// Final prompt after refinement on validation set`
**7 for** *each batch $\mathcal{I}_j$ ($j = 1$ to $k$)* **do**
**8**     Generate final annotations:
**9**     $\mathcal{T}_j \leftarrow \texttt{GPT4V}(\mathcal{I}_j, P^*)$;
**10 return** $P^*$, $\{\mathcal{T}_1, \ldots, \mathcal{T}_k\}$

---

**Algorithm 2:** Occlusion-Aware Gaze Point Aggregation

---

**Input:** Image sequence $\{I_\tau\}$ for $\tau \in [t - \delta, t]$,
      Gaze points $\{g_\tau\}$ at each timestamp $\tau$,
      Optical flow model (RAFT),
      Thresholds $\epsilon = 20$, $\eta = 0.60$
**Output:** Aggregated heatmap $H_t$

**1** Initialize $H_t$ as empty
**2 for** $\tau = t - \delta$ **to** $t$ **do**
**3**     Compute forward flow $f_{\tau \to t}$ and backward flow $f_{t \to \tau}$;
**4**     **for** *pixel $\mathbf{p}$ in $I_\tau$* **do**
**5**        $\hat{\mathbf{p}} = \mathbf{p} + f_{\tau \to t}(\mathbf{p})$;
**6**        $\Delta = \|f_{\tau \to t}(\mathbf{p}) + f_{t \to \tau}(\hat{\mathbf{p}})\|$;
**7**        Mark $\mathbf{p}$ as **occluded** if $\Delta > \epsilon$;
**8**     Compute occlusion ratio: $\eta_{\text{observed}} = \frac{1}{H \times W} \sum \mathbf{1}(\Delta > \epsilon)$;
**9**     **if** $\eta_{observed} > \eta$ **then**
**10**        Discard gaze point $g_\tau$ (major occlusion);
**11**     **else**
**12**        Translate $g_\tau$ to $g_{\tau \to t}$ using $f_{\tau \to t}$;
**13**        Add Gaussian heatmap from $g_{\tau \to t}$ to $H_t$;
**14** Normalize $H_t$;
**15 return** $H_t$

---

### A.4 Model Overview and Components

In our study, we initially employ the open-source version of the Flamingo model (Awadalla et al., 2023), based on the original Flamingo architecture proposed by Alayrac et al. (2022). Flamingo is a VLM designed to process interleaved image-text inputs, featuring a pre-trained vision encoder, a trainable Perceiver Resampler for constructing fixed-length visual tokens, and a cross-attention

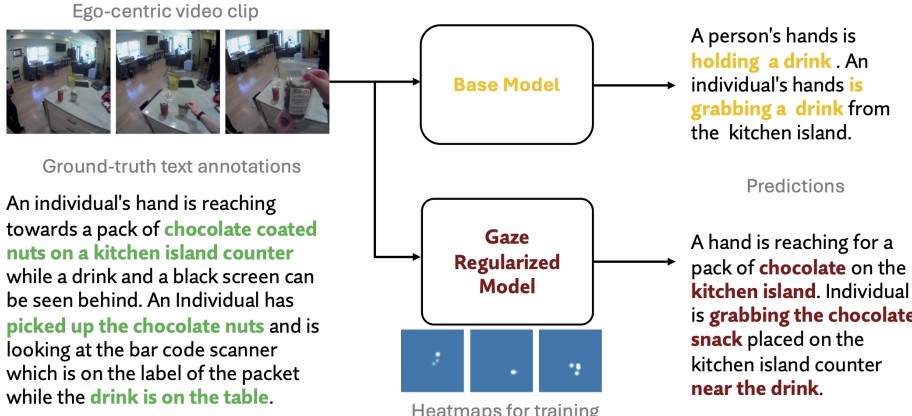

Figure 6: **Simple Illustration of Activity Understanding.** The model receives a sequence of egocentric frames as input and generates a fine-grained prediction of the current activity. While the base model misidentifies the object being picked up ('a drink'), the gaze-regularized model correctly predicts 'chocolate' by leveraging attention aligned with human gaze during training. Importantly, spatial gaze heatmaps are used only to guide attention during training and are not used at inference. Predicted annotations are shown on the right, with ground-truth descriptions below.

language decoder. To extend its capabilities for egocentric video understanding, we introduce a gaze-regularized attention mechanism that is placed before the Perceiver Resampler, enabling alignment between model attention and human gaze distributions.

Building on this design, we generalize our gaze-regularization framework to other transformer based VLMs, including LLaVA, InternVL, and LaViLa, by modifying their input pipelines as required to accept a sequence of image frames instead of a single frame. Our core idea remains consistent across models: a gaze-based attention regularization mechanism is applied prior to fusing visual tokens with the language decoder, modulating attention to emphasize gaze-informed spatial regions.

It is important to note that while we adopt the architectures of models like LLaVA and InternVL, we do not use their original pretraining paradigms or loss functions. As such, the performance reported for these models should not be interpreted as a ranking of their full potential. Rather, our intention is to demonstrate the generalizability and impact of gaze regularization on different VLM architectures under a consistent training setup.

Inspired in part by the transformer-based egocentric gaze modeling approach of Lai et al. (2024), our method uses gaze heatmaps only during training to guide attention. During inference, the models rely solely on RGB image inputs, making them suitable for deployment scenarios where gaze data is unavailable. A simple illustration of this setup, along with example outputs for activity understanding, is shown in Figure 6.

To further ensure robustness, we incorporate an occlusion-check module based on optical flow consistency that filters unreliable gaze points during temporal aggregation. This improves the fidelity of gaze supervision and ensures that only visually relevant regions guide the model's attention. The resulting framework enables gaze-guided fine-grained predictions and current activity understanding across multiple VLMs, while maintaining modularity and scalability.

## A.5  Other Experiments

In this section, we present a series of ablation studies that explore the core design choices in our gaze-regularized framework. We begin by defining the evaluation metrics used to measure semantic alignment between predicted and ground-truth activity descriptions. Next, we conduct a sanity-check experiment to evaluate whether existing pretrained VLMs are capable of future activity prediction in egocentric settings.

We then analyze how the size and temporal density of gaze points affect model performance by varying size and number of gaze points used for heatmap aggregation. To disentangle performance

gains due to architecture versus gaze supervision, we augment the base model with additional self-attention layers but without gaze regularization. We also test whether providing gaze coordinates in textual form (rather than as spatial heatmaps) yields comparable results.

Finally, we examine the effect of the observation horizon. Most of the ablations are conducted using the OpenFlamingo architecture as the base model , as it was the model initially used to get some preliminary results and intuition about how to proceed further. Once the framework was validated, we generalized it to other architectures, as discussed in the main paper.

### A.5.1 Architectures Used

We tested our method using a set of established open-source models, including OpenFlamingo, LaVila, InternVL 2.5-1B, and OpenLLaVA (LLaVA-NeXT-Vicuna-7B). These models provided a strong and diverse foundation for our evaluation. Their clear documentation and straightforward implementation made them practical choices for our experiments.

### A.5.2 Evaluation Metrics

For evaluation, we propose using a semantic transformer (Reimers and Gurevych, 2019) to provide a quantifiable score that compares the generated output with the ground-truth action text, along with the ROUGE-L scores for some tables. This scoring system is designed to ensure that the model is not penalized for semantically equivalent phrasings. For example, 'Football is being played by people' and 'People are playing football' convey the same meaning and should yield similar scores. At the same time, the system should penalize outputs with nonsensical or incoherent word order. While some ablation tables in the appendix report both semantic and ROUGE-L scores for completeness, we omitted ROUGE-L from the main paper due to space constraints and because it exhibited highly correlated trends with the semantic scores. This allowed us to present a more concise evaluation while still conveying the key performance improvements.

### A.5.3 Evaluating Pretrained VLMs for Future Activity Prediction

Pretrained VLMs such as InternVL and OpenFlamingo have demonstrated impressive performance on a variety of tasks, including visual question answering, image captioning, and multi-modal reasoning. However, these models have not been explicitly designed or optimized for future activity prediction. To assess whether such models can be directly applied to short-horizon future prediction, we conduct a sanity-check experiment: we provide each VLM with a sequence of image frames and evaluate its ability to generate predictions about future actions.

Table 8: Evaluation on pretrained VLMs for activity prediction

| MODEL | SEMANTIC SCORE ($\uparrow$) |
|---|---|
| INTERNVL-2-1B | 0.1572 |
| INTERNVL-2-2B | 0.1601 |
| INTERNVL-2.5-1B | 0.1596 |
| OPENFLAMINGO-3B | 0.1810 |
| OPENFLAMINGO-4B | 0.1878 |
| **OUR METHOD** | **0.7505** |

While this evaluation is not entirely fair,given differences in training objectives, data domains, and task formulation ,it offers a useful diagnostic of how well existing models transfer to our setup. As shown in Table 8, current VLMs exhibit limited performance on our test set. This may stem from domain shift or the lack of temporal modeling in their pretraining. Nonetheless, the results highlight a promising opportunity for future work to extend VLM capabilities toward temporally grounded tasks like short-horizon future activity prediction.

### A.5.4 Impact of Gaussian Overlay Size in Singular vs. Aggregated Gaze Models

In our exploration of gaze-regularized models, we investigated two primary strategies for representing gaze: (i) the *singular gaze model*, which uses a single gaze point per frame to form a spatial heatmap,

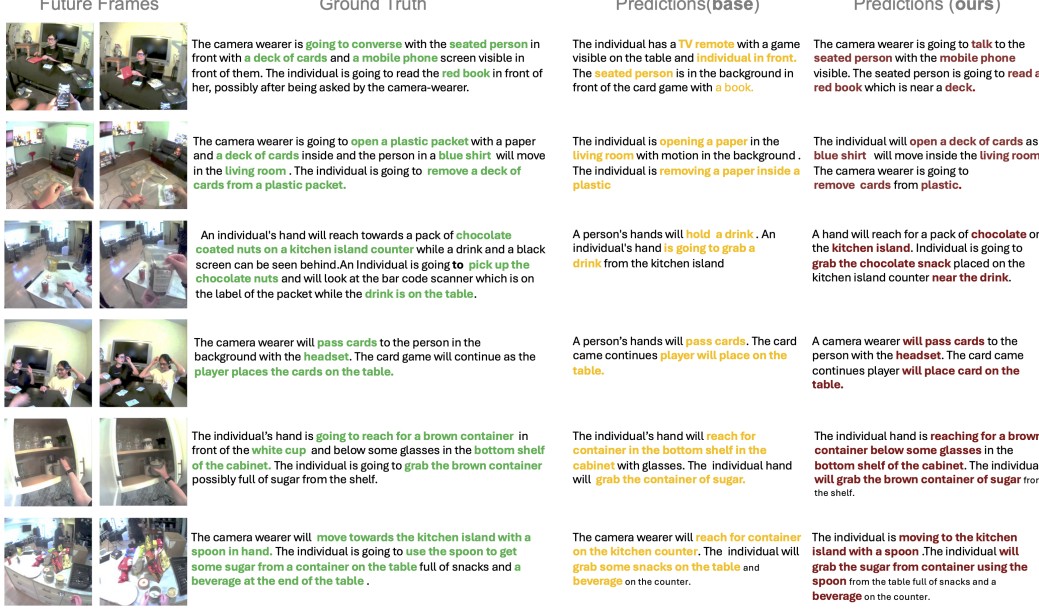

| Future Frames | Ground Truth | Predictions(**base**) | Predictions (**ours**) |
|---|---|---|---|
| | The camera wearer is going to converse with the seated person in front with a deck of cards and a mobile phone screen visible in front of them. The individual is going to read the red book in front of her, possibly after being asked by the camera-wearer. | The individual has a TV remote with a game visible on the table and individual in front. The seated person is in the background in front of the card game with a book. | The camera wearer is going to talk to the seated person with the mobile phone visible. The seated person is going to read a red book which is near a deck. |
| | The camera wearer is going to open a plastic packet with a paper and a deck of cards inside and the person in a blue shirt will move in the living room. The individual is going to remove a deck of cards from a plastic packet. | The individual is opening a paper in the living room with motion in the background. The individual is removing a paper inside a plastic | The individual will open a deck of cards as blue shirt will move inside the living room. The camera wearer is going to remove cards from plastic. |
| | An individual's hand will reach towards a pack of chocolate coated nuts on a kitchen island counter while a drink and a black screen can be seen behind.An Individual is going to pick up the chocolate nuts and will look at the bar code scanner which is on the label of the packet while the drink is on the table. | A person's hands will hold a drink. An individual's hand is going to grab a drink from the kitchen island | A hand will reach for a pack of chocolate on the kitchen island. Individual is going to grab the chocolate snack placed on the kitchen island counter near the drink. |
| | The camera wearer will pass cards to the person in the background with the headset. The card game will continue as the player places the cards on the table. | A person's hands will pass cards. The card came continues player will place on the table. | A camera wearer will pass cards to the person with the headset. The card came continues player will place card on the table. |
| | The individual's hand is going to reach for a brown container in front of the white cup and below some glasses in the bottom shelf of the cabinet. The individual is going to grab the brown container possibly full of sugar from the shelf. | The individual's hand will reach for container in the bottom shelf in the cabinet with glasses. The individual hand will grab the container of sugar. | The individual hand is reaching for a brown container below some glasses in the bottom shelf of the cabinet. The individual will grab the brown container of sugar from the shelf. |
| | The camera wearer will move towards the kitchen island with a spoon in hand. The individual is going to use the spoon to get some sugar from a container on the table full of snacks and a beverage at the end of the table. | The camera wearer will reach for container on the kitchen counter. The individual will grab some snacks on the table and beverage on the counter. | The individual is moving to the kitchen island with a spoon .The individual will grab the sugar from container using the spoon from the table full of snacks and a beverage on the counter. |

Figure 7: **Extended Qualitative Results of Future Activity Prediction.** Examples comparing predictions from the base model and our gaze-regularized model against ground-truth annotations and actual future frames. The gaze-regularized model correctly predicts specific objects and actions (e.g., "picking up chocolate") while the base model makes less accurate predictions (e.g., incorrectly predicting "a drink"). In addition, the gaze regularized model also gives finer-grained predictions by correctly predicting the color (e.g., "brown container"), objects surrounding the main object (e.g., "below some glasses"), which the base model fails to do in the predictions. By highlighting such properties, giving instructions to other humans or machines becomes easier, especially if there are multiple such objects involved. Key words in the predictions are highlighted to emphasize differences in prediction specificity and accuracy.

and (ii) the *aggregated gaze model*, which combines gaze points from a temporal window with occlusion filtering.

To assess the role of spatial spread in these representations, we varied the standard deviation ($\sigma$) of the Gaussian kernel used to generate the gaze heatmap for both models. In the singular gaze model, increasing $\sigma$ which expands the influence region around each gaze point , led to modest performance improvements. This suggests that slightly broadening the gaze-induced attention might help the model focus on semantically meaningful areas rather than overly localized pixels.

In contrast, increasing the Gaussian smoothing in the aggregated gaze model had minimal impact. As shown in Table 9, its performance remained nearly unchanged. This indicates that the strength of the aggregated model stems from its temporal diversity and occlusion-aware gaze selection, rather than from the spatial extent of individual gaze contributions.

These experiments were initially conducted using the OpenFlamingo architecture, which served as our primary reference during early ablation studies.

Overall, these findings highlight an important distinction: while spatial smoothing can benefit isolated gaze representations, temporally aggregated gaze signals already encode spatial redundancy via multiple temporally aligned points. Future work may investigate how the interplay between Gaussian spread ($\sigma$), image resolution, and aggregation window size affects performance.

### A.5.5   Human Evaluation of Visual Hallucinations

To assess whether our gaze regularization method mitigates visual hallucination in VLMs, we conducted a controlled human evaluation. This analysis was motivated by the fact that standard

Table 9: Effect of the size of the gaze point overlays on model accuracy

| MODEL | $\sigma$ | SCORE(↑) | F-SCORE |
|---|---|---|---|
| SINGULAR | 20 | 0.6211 | 0.4327 |
| SINGULAR (LARGER OVERLAYS) | 50 | 0.6512 | 0.4463 |
| AGGREGATED | 20 | **0.7505** | **0.5173** |
| AGGREGATED (LARGER OVERLAYS) | 50 | 0.7436 | 0.5034 |

hallucination benchmarks (e.g., HallusionBench (Guan et al., 2024)) do not contain gaze annotations or surrogate attention signals, making direct application of our method infeasible on these benchmarks.

We randomly selected 200 examples from our test set and generated textual outputs using both the base model and our gaze-regularized model. These responses were then randomly shuffled and presented to a human evaluator who was blind to the model source. For each example, the evaluator was shown the corresponding ground-truth video context and annotations, enabling reliable identification of hallucinated content.

The evaluator was instructed to flag any instance where the model's output mentioned objects or actions that were either:

1. Clearly absent from the visual input, or

2. Unrelated to the actual scene content

Following the evaluation protocol from HallusionBench, we computed the $C_I$ score, defined as the ratio of annotations with clearly hallucinated instances to the total number of annotations:

Table 10: Human evaluation results for visual hallucination. Our gaze-regularized model shows a significant reduction in hallucinated content.

| Model Variant | Hallucinated Cases | Total Samples | $C_I$ |
|---|---|---|---|
| Base Model | 41 | 200 | 0.205 |
| Gaze-Regularized Model | 28 | 200 | 0.140 |

This improvement suggests that aligning the model's attention with human gaze patterns helps ground the language generation in visually present content, reducing the tendency to hallucinate objects or actions not supported by the visual evidence. While this evaluation was conducted only on our dataset, the findings indicate the potential of gaze-guided training for addressing hallucination in VLMs more broadly. Future work could explore generating proxy gaze signals to enable evaluation on standard hallucination benchmarks.

### A.5.6 Impact of Using More Gaze Points in Temporal Aggregation

To explore the impact of using a larger number of points (or a longer aggregation duration), we trained an aggregated gaze model with the openflamingo architecture utilizing 12 frames instead of 6 (and hence the aggregation time $\delta$ is 400 ms). The results of this experiment are provided in Table. 11. This minimal improvement is likely due to the occlusion checks already filtering out less informative points, reducing the marginal utility of including more frames and limiting the number of usable gaze points. In cases with minimal occlusion, the slight decrease in performance can be compared to that observed in an aggregated gaze model with larger overlays, where performance also decreased slightly. This result suggests that maybe utilizing an excessive number of gaze points could confuse the model. However, as this explanation is currently based on intuition rather than extensive analysis, this experiment was not included in the main paper.

Table 11: Comparison when more aggregated points are used for the gaze-regularized model

| GAZE POINTS | SCORE(↑) | F-SCORE(↑) |
|---|---|---|
| 6 | **0.7505** | **0.5173** |
| 12 | 0.7398 | 0.4971 |

### A.5.7 Impact of Occlusion Filtering on Model Performance

In the aggregated gaze model, gaze points are collected within a specified time interval $\delta$ (200 ms). However, in dynamic environments, aggregating gaze points from the interval $[t - \delta, t]$ may introduce inaccuracies due to changes in the scene or camera movement. To mitigate this, we introduced an occlusion check to ensure that only relevant gaze points are aggregated. More details about the occlusion check method can be found in Sec. A.3 in the Appendix. To evaluate the impact of this adjustment, we conducted experiments comparing models with and without the occlusion check in the future prediction task. As shown in Table 12, the openflamingo implementation of our framework incorporating the occlusion check slightly outperforms the one without it. The difference in the evaluation metrics can be attributed to the fact that only relevant and accurate gaze points are considered, which reduces noise and prevents the model from being confused by irrelevant data.

Table 12: Effect of using occlusion filtering during training

| MODEL | SCORE | F-SCORE ($\uparrow$) |
|---|---|---|
| GAZE-REG (W/O OCCL.) | 0.7298 | 0.4800 |
| GAZE-REG (W/ OCCL.) | **0.7505** | **0.5173** |

### A.5.8 Can Gaze Coordinates alone improve Model Performance?

In our studies, we converted gaze data from coordinate text form into heatmaps, which were then utilised by our gaze regularizer during training time. This transformation allows the regularizer to highlight important visual regions and aligns more closely with how humans perceive spatial information (Laeng et al., 2014). To conduct a sanity check and assess whether using gaze data in visual form is more suitable than using gaze in text form , we trained a gaze-regularized model that utilizes gaze coordinates as text input. Our results indicated that using gaze data in text form improved performance compared to the base model without gaze. However, it still fell short of the performance achieved by our model (as shown in Table 13). This highlights the importance of not just including gaze data, but representing it in a specific format such that it can be used aligned to modulate the spatial nature of model attention.

Table 13: Effect of using gaze information in text form and comparison with other models

| MODEL | SCORE($\uparrow$) | F-SCORE ($\uparrow$) |
|---|---|---|
| BASE | 0.6525 | 0.4318 |
| AGGREGATED GAZE(IN TEXT FORM) | 0.7021 | 0.4630 |
| AGGREGATED GAZE | **0.7505** | **0.5405** |

### A.5.9 Addition of Self Attention Blocks to the Base Model

To ensure a fair comparison, we evaluate whether the performance improvements in our gaze-regularized model stem solely from its architectural enhancements , specifically, the extra attention layers that operate on a global query derived from the full image sequence. To test this, we augment the base OpenFlamingo model by adding two self-attention layers and providing it with the same global query, but without applying any gaze-based regularization. This allows us to isolate the effect of architectural changes from the influence of gaze supervision. As shown in Table 14, the modified base model does show improved performance compared to the original baseline. However, it still falls short of the performance achieved by the gaze-regularized models. This suggests that while incorporating global queries and self-attention blocks enhances the model's capacity to aggregate contextual information, it is the gaze-guided regularization that ultimately drives stronger alignment with human attention and increases model performance.

### A.5.10 Training Base VLM with Gaze-Embedded Images

As a sanity check, we evaluate what happens when baseline models are trained on gaze-embedded RGB images, where gaze heatmaps are directly superimposed onto the original RGB frames. This

Table 14: Comparison of base model with attention block against gaze regularized models

| MODEL | SCORE(↑) | F-SCORE(↑) |
|---|---|---|
| BASE | 0.6525 | 0.4318 |
| BASE (W SELF-ATTENTION) | 0.6701 | 0.4393 |
| GAZE-REGULARIZED | **0.7505** | **0.5173** |

setup allows us to assess whether performance improvements could arise simply from exposing the model to gaze-like visual cues, without any explicit use of gaze during training or inference,as is the case in our gaze-regularized framework, which does not take gaze as input. All models were trained under the same conditions, using identical data splits. As shown in Table 15, using gaze-embedded inputs yields a modest performance gain over standard RGB inputs. However, this improvement remains well below the performance achieved by our gaze-regularized models. These results suggest that the benefits of our approach stem from the explicit use of gaze as a training-time regularizer, rather than from simply incorporating gaze like patterns in the input images.

Table 15: Comparison for egocentric event prediction when gaze-overlaid images are provided to base VLM.

| MODEL | SCORE(↑) | F-SCORE |
|---|---|---|
| BASE W RGB IMAGE | 0.6525 | 0.4318 |
| BASE W GAZE-EMBEDDED IMAGE | 0.6873 | 0.4435 |
| GAZE REGULARIZED | **0.7505** | **0.5173** |

### A.5.11  Effect of Temporal Gaze Aggregation on Performance

As described in the main paper, we construct gaze heatmaps by first transforming individual gaze points into spatial maps and then aggregating them over a short temporal window. This aggregation is accompanied by occlusion filtering to ensure that only visually valid gaze points contribute to the final heatmap used for attention regularization.

To understand the specific contribution of temporal aggregation, we compare performance against a baseline that uses only a single gaze point at each timestep–referred to as the *singular gaze model*. In this case, the gaze heatmap $H_t$ at time $t$ is defined as:

$$H_t = m_t = \pi(G_\sigma * \mathbf{1}(g_t)), \tag{11}$$

is an indicator function with value 1 at position $g_t$ and 0 elsewhere, $G_\sigma$ is a Gaussian kernel with standard deviation $\sigma$, $*$ denotes convolution, and $\pi$ represents a normalization so the heatmap sums to 1.

Our experiments show that aggregating gaze points over a short interval (e.g., 200 milliseconds) consistently improves performance over the singular gaze variant. This improvement arises because temporal aggregation captures the stable and temporal structure of visual attention across frames,whereas single gaze points can be often noisy, particularly if sampled during saccades or fleeting fixations. In addition, over-regularization of attention to a small space (pertaining to a singular overlaid point) might not always be beneficial, as shown by the performance degradation as compared to the baseline model where no gaze regularization is used.

By integrating gaze information from multiple frames (and by including an occlusion filtering mechanism), the model gains a richer and more reliable supervisory signal that better aligns its attention with human intent. This temporal consistency enables more accurate understanding of ongoing actions and better prediction of upcoming behavior. A simplified illustration comparing singular and temporally aggregated gaze heatmaps is shown in Figure 8.

### A.5.12  Change in Observation Horizon for Future Prediction

In the main paper, we reported results for future prediction tasks using an anticipation horizon of $\tau_a = 2$ seconds and an observation horizon of $\tau_o = 5$ seconds. To further analyze the model's behavior, we

Table 16: Semantic performance on future prediction and activity understanding tasks comparing singular and aggregated gaze models. Gains indicate the absolute improvement from using aggregated gaze.

| MODEL | FUTURE PREDICTION | | | ACTIVITY UNDERSTANDING | | |
|---|---|---|---|---|---|---|
| | SINGLE | AGGREGATED | GAIN | SINGLE | AGGREGATED | GAIN |
| OPENFLAMINGO | 0.6512 | **0.7505** | +9.9% | 0.6913 | **0.7848** | +9.4% |
| MODIFIED OPENFLAMINGO | 0.6024 | **0.7200** | +11.8% | 0.6631 | **0.7300** | +6.7% |
| LAVILA NARRATOR | 0.6037 | **0.6924** | +8.9% | 0.6364 | **0.7268** | +9.0% |
| INTERNVL | 0.6223 | **0.7318** | +10.9% | 0.6813 | **0.7404** | +5.9% |
| OPENLLAVA | 0.5814 | **0.6771** | +9.6% | 0.6500 | **0.7027** | +5.3% |

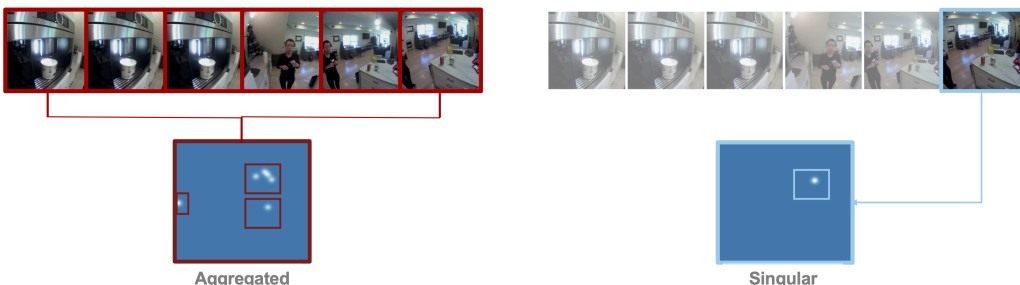

Figure 8: **Illustration of singular and aggregated gaze heatmaps.** For the *singular gaze* heatmap, only the gaze point associated with the final heatmap is utilized. On the other hand, for the *aggregated gaze* heatmap, gaze points over an interval $\delta = 200ms$ are collected to form the aggregated heatmap.

conducted an additional experiment by reducing the observation horizon to $\tau_o = 3$ seconds i.e., using a shorter input sequence to predict the same future window. Interestingly, this reduced observation window led to slightly improved performance for some models, while maintaining consistent overall gains when gaze regularization is used as opposed to a baseline model, as shown in Table 17.

This suggests that even with limited visual context, our gaze-regularized models can effectively anticipate future actions. Interestingly, while humans leverage visual working memory for short-term planning, this memory may not be effectively simulated over longer temporal horizons in our current framework. As a result, tracking the visual cues may become harder as the anticipation window grows, leading to reduced predictive accuracy. While this remains a speculative explanation, it motivates further investigation into integrating memory inspired mechanisms into our gaze-inspired framework. Accordingly, we present these exploratory results in the appendix rather than in the main paper as the explanation is intuitive and not experimentally investigated.

Table 17: Semantic performance on future prediction task with reduced observation horizon (3 seconds). The aggregated gaze model continues to outperform the base model across all architectures.

| MODEL | SINGLE | AGGREGATED | GAIN |
|---|---|---|---|
| OPENFLAMINGO | 0.6616 | **0.7565** | +9.5% |
| MODIFIED OPENFLAMINGO | 0.6200 | **0.7281** | +10.8% |
| LAVILA NARRATOR | 0.6144 | **0.6966** | +8.2% |
| INTERNVL | 0.6301 | **0.7279** | +9.8% |
| OPENLLAVA | 0.5851 | **0.6731** | +8.8% |

### A.5.13 Activity Understanding for Static Image

We examine whether a static scene, independent of sequential context, is sufficient for the gaze-regularized models to understand an ongoing activity, noting that VLMs are often used in static-scene setting. While a single image lacks temporal context, it contains visual cues and object affordances that hint at probable actions. Unlike future prediction, which benefits from evolving gaze patterns, static scene understanding relies mainly on visible objects. If gaze is beneficial, it should highlight key

regions that aid activity understanding. To test this, we compare the base and gaze-regularized models in generating fine-grained descriptions from single images, isolating the impact of gaze without temporal cues. Results from Table 18 show that the gaze-regularized model achieves a modest 1.3 % improvement over the base model, indicating that gaze has moderate impact on scene understanding in single-image settings. However, its full potential is realized in tasks where evolving gaze patterns provide richer contextual cues. Hence for the activity understanding tasks in the main paper, we provide a long enough observation horizon $\tau_o = 3s$ such that the activity can be understood, as well as to utilize the evolving temporal gaze patterns.

Table 18: Comparison between base model and gaze-regularized models (with different regularization coefficients) for activity understanding tasks.

| MODEL | SCORE | F-SCORE |
|---|---|---|
| BASE | 0.6897 | 0.4432 |
| GAZE-REGULARIZED | **0.7014** | **0.4611** |

### A.5.14 Effect of Using Sequence-Level vs. Frame-Level Queries for Attention Computation

Table 19: Impact of using sequence-level queries on semantic prediction for activity understanding. Using global queries from the entire input sequence improves performance on both in-distribution and out-of-distribution data.

| MODEL VARIANT | SCORE (TEST) | SCORE (OOD) |
|---|---|---|
| GAZE-REGULARIZED (W/O GLOBAL QUERY) | 0.7284 | 0.6902 |
| GAZE-REGULARIZED (WITH GLOBAL QUERY) | **0.7505** | **0.7305** |

In our main framework, we compute a global query using the entire input image sequence, rather than generating separate queries for each individual frame. This choice supports better temporal generalization. If frame-level queries are used, the model processes each image independently, without awareness of the broader activity context. As a result, the attention mechanism may become overly dependent on visual patterns specific to individual frames.

This becomes particularly problematic in tasks like future prediction, where the same frame could occur in different activity sequences. Without access to temporal context, a frame-level query cannot distinguish between these scenarios, increasing the risk that the model memorizes attention patterns instead of also relying learning task-relevant cues. In contrast, a sequence-level (global) query captures the dynamics of the entire clip, encouraging the model to focus on features that reflect the underlying activity rather than frame-specific objects. From Table 19, the results support our intuition: incorporating global queries not only improves semantic scores but also results in a smaller performance drop on out-of-distribution data compared to using frame-wise queries.

### A.5.15 Generalization to Other Egocentric Tasks

**Challenges with Native Ego4D Benchmarks**   Our method requires gaze supervision during training, which limits us to the subset of Ego4D videos containing gaze annotations. While Ego4D provides extensive benchmarks for activity classification, video-text retrieval, and forecasting, these benchmarks primarily use videos *without* gaze data. Furthermore, the gaze-annotated subset lacks the structured labels required for these benchmarks (e.g., verb/noun labels, bounding boxes, or precise temporal segments). This inherent dataset limitation prevented direct application of our method to standard Ego4D benchmarks. However, we provide a couple of additional experiments below to highlight the efficacy of our method, by using a pseudo-gaze or a predicted gaze signals (using a third-party pre-trained gaze prediction module) even for the videos without gaze but the other structured labels, as well as a coarse narration prediction task- utilizing the original coarse narrations present in the subset for gaze data to show the improvements made by the gaze regularized model over the base model.

**Coarse Narration Prediction Task**   In this experiment, we leveraged the narrations available in the gaze-annotated subset. These narrations provide coarse, first-person descriptions of activities

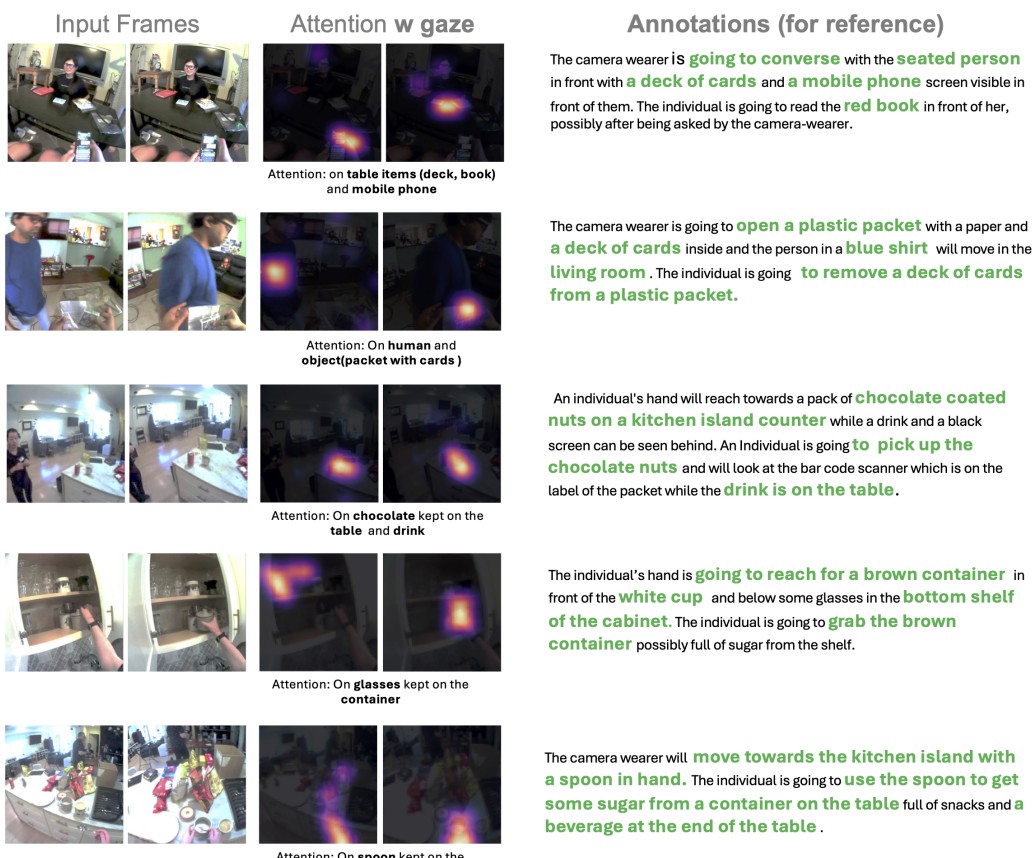

| Input Frames | Attention **w gaze** | Annotations (for reference) |

The camera wearer **is going to converse** with the **seated person** in front with **a deck of cards** and **a mobile phone** screen visible in front of them. The individual is going to read the **red book** in front of her, possibly after being asked by the camera-wearer.

Attention: on **table items (deck, book)** and **mobile phone**

The camera wearer is going to **open a plastic packet** with a paper and **a deck of cards** inside and the person in a **blue shirt** will move in the **living room**. The individual is going **to remove a deck of cards from a plastic packet.**

Attention: On **human** and **object(packet with cards )**

An individual's hand will reach towards a pack of **chocolate coated nuts on a kitchen island counter** while a drink and a black screen can be seen behind. An Individual is going **to pick up the chocolate nuts** and will look at the bar code scanner which is on the label of the packet while the **drink is on the table**.

Attention: On **chocolate** kept on the **table** and **drink**

The individual's hand is **going to reach for a brown container** in front of the **white cup** and below some glasses in the **bottom shelf of the cabinet**. The individual is going to **grab the brown container** possibly full of sugar from the shelf.

Attention: On **glasses** kept on the **container**

The camera wearer will **move towards the kitchen island with a spoon in hand.** The individual is going to **use the spoon to get some sugar from a container on the table** full of snacks and **a beverage at the end of the table** .

Attention: On **spoon** kept on the **container** and **drink**

Figure 9: **Extended Comparison of Attention Maps.** This figure compares attention distributions from the base model (without gaze regularization) and our gaze-regularized model. Our method produces attention maps that are more semantically aligned with human gaze, leading to better focus on task-relevant regions. For instance, in the second-to-last row, the model attends to both the glasses and the container–objects emphasized in the ground truth annotation. In the final row, attention is correctly directed toward the hand and the container, improving the model's ability to interpret and predict the ongoing activity.

recorded by camera wearers. We defined a **coarse future narration prediction** task to evaluate whether our approach generalizes to scenarios with less detailed supervision.

As shown in Table 20, our gaze-regularized model achieves modest but consistent improvements over the baseline, with a 3.9% increase in semantic score and 2.2% in F-score. The smaller gains compared to fine-grained prediction tasks (which showed 7-10% improvements) support our core hypothesis: gaze supervision is particularly effective for capturing subtle spatial and object details that coarse labels often overlook.

Table 20: Performance on coarse future narration prediction task. Gains are modest compared to fine-grained tasks, supporting our claim that gaze helps capture subtle details.

| Model Variant | Semantic Score ↑ | F-Score ↑ |
|---|---|---|
| Base Model | 0.7232 | 0.4982 |
| Gaze-Regularized Model | **0.7621** | **0.5200** |

**Generalization with Predicted Gaze** To test our method's applicability to standard benchmarks without human gaze, we applied it to the Ego4D Short-Term Object Interaction Anticipation (STA) benchmark using *predicted* gaze heatmaps. We employed an off-the-shelf gaze prediction model to

generate pseudo-gaze signals, which were then used for attention regularization during training via our standard KL-divergence loss.

As shown in Table 21, our approach improves performance on this established benchmark, with absolute gains of 2.88 mAP for noun prediction and 2.16 mAP for verb+noun prediction. These results demonstrate that our gaze regularization framework can generalize to tasks without human gaze annotations, suggesting that even approximate attention signals can provide beneficial supervision.

Table 21: Results on Ego4D Short-Term Anticipation (STA) benchmark using predicted gaze. Our method shows consistent improvements despite using surrogate gaze signals.

| Model Variant | mAP (Noun) ↑ | mAP (Noun + Verb) ↑ |
|---|---|---|
| Base Model | 22.13 | 11.29 |
| Gaze-Regularized Model | **25.01** | **13.45** |

These experiments collectively demonstrate the versatility of our approach: it provides substantial benefits for fine-grained understanding tasks with detailed annotations, modest improvements for coarse prediction tasks, and remains effective even when using predicted gaze signals on standard benchmarks.

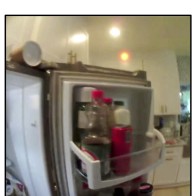 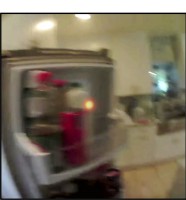 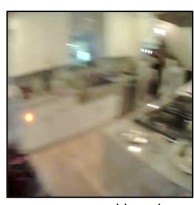

**Our Model**
Camera wearer appears blurry. Image is moving fast and blurred.

**Ground Truth**
Camera wearer is near a refrigerator which contains items, most probably a ketchup bottle. The scene appears to be moving fast and blurred.

(a) Failure in current activity understanding if majority of the input sequences are blurred.

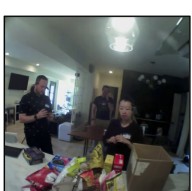 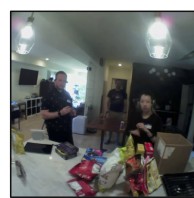 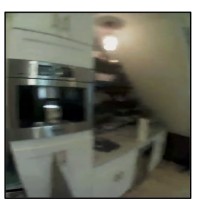

**Ground Truth (Future Prediction)**
Camera wearer is going to talk to people and look at the snacks on the white table counter. Camera wearer will move to a blurred scene which looks like a kitchen with an oven.

(b) Example of ground truth annotations containing the word 'blur' which is not useful in real life deployment for prediction and hence had to be removed.

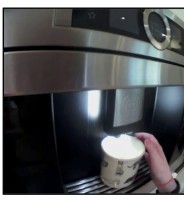 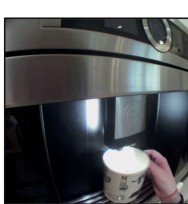 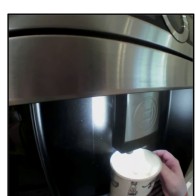

**Ground Truth**
Camera wearer is placing a white cup waiting for a beverage. Camera wearer has placed a white cup under a coffee machine for beverage. Camera wearer is still waiting with a white cup placed under a stainless-steel coffee machine

(c) Potential examples where even during a 3 second interval, the actions do not change drastically and hence the observational cues are not as effective.

Figure 10: **Example failure case and qualitative observations.** *Top:* When the majority of frames in the input sequence are significantly blurred, even gaze-regularized models struggle to produce accurate predictions. Although the ground-truth annotations preserve contextual intent, the lack of clear visual input limits the model's ability to reason about ongoing actions. *Middle:* Examples of image-text pairs that were excluded during dataset construction because they contained words like "blur" or "unclear," which offer little utility for activity understanding or future prediction. We aimed to minimize the inclusion of such samples. *Bottom:* A case where the actions remain largely consistent across the sequence, making the activity easier to infer regardless of the modeling approach.

**Training and Evaluation details**    Both the base model and the gaze-regularized model were trained using two NVIDIA A800 80GB GPU cards. For initial experiments, we used the OpenFlamingo architecture to develop and evaluate our approach. The base OpenFlamingo model required approximately 36–38 hours to train, while the corresponding gaze-regularized version took around 50 hours.

This additional time is due to the added regularization computation and the insertion of gaze-aligned attention blocks. Training was conducted with a batch size of 32 and a learning rate of $7 \times 10^{-5}$ over 10 epochs. The vision encoders were kept frozen and pre-trained. To accelerate training, we employed Fully Sharded Data Parallel (FSDP), which efficiently distributes model parameters and gradients across GPUs, improving memory usage and speed. Data loading was managed using the WebDataset loader, with datasets converted to .tar format for compatibility with both WebDataset and FSDP. After validating our method with OpenFlamingo, we extended the same training pipeline to other architectures such as InternVL, LaViLa, and OpenLLaVA. In each case, the integration of our gaze-regularized component followed the same principle: it was inserted immediately after the visual encoder and before the language decoder, allowing for modular modulation of attention without disrupting the rest of the model architecture.

To give a comparison in terms of model complexity, the base model for openflamingo has approximately 944 million parameters. The gaze-regularized model with 2 additional attention blocks, which is the best performing model, has approximately 966 million parameters. For evaluation, models were assessed using the semantic transformer (SBERT) developed by Reimers and Gurevych (2019), along with ROUGE-L scores. Regarding the runtime, we would like to clarify that the evaluation was conducted using RGB images for the base model and the gaze-regularized model. On average, the base model took approximately 1.7-1.8 seconds to process a sequence while the latter took 2.2 - 2.3 seconds . However, this runtime does not include the time taken for image loading or pre-processing overhead. The testing was also conducted on a NVIDIA A800 80GB GPU card.

### A.6 Discussion and Limitations

In the following section, we briefly discuss some limitations of our current framework and outline promising directions for future work.

While our gaze-regularized framework offers strong improvements in egocentric understanding tasks, certain constraints remain. To maintain computational efficiency, we downsample videos to 1 frame per second. Although this preserves high-level temporal patterns, it may limit sensitivity to rapid or transient actions. Nonetheless, we find that the current performance and predictions capture egocentric behavior understanding in majority of the scenarios.

A common challenge in egocentric vision is the presence of motion blur or poor visual quality. In sequences where most frames are heavily blurred, even gaze-regularized models struggle to generate accurate predictions, as both visual and gaze cues become less informative. Figure 10 illustrates such a case. Even though we tried to identify most of such ground truth annotations and mitigate them, some cases still persist and this is one of the limitations.

While our occlusion-aware gaze filtering improves robustness, the reliability of gaze supervision still depends on accurate calibration. Misalignment in eye-tracking data can lead to suboptimal heatmaps, which in turn may misguide the attention regularizer. For training the models from scratch using custom data, it is necessary to ensure that the gaze calibration used for collecting training data is in tune with the camera wearer. Our findings offer encouragement and also point towards the need for deeper exploration of how gaze aligns with visual representations in VLMs. For example, recent observations in Bolya et al. (2025) suggest that the final-layer features of vision encoders may not always be ideal for all downstream tasks. This raises an interesting parallel: just as feature selection matters in model design, it is worth asking whether certain stages of gaze processing e.g., early fixations vs. late context scanning ,align better with specific types of reasoning, such as classification, visual search, or question answering.

This leads to a promising direction for interdisciplinary collaboration. Psychological research has long shown that gaze behavior is task-dependent: humans scan scenes differently when searching for an object versus answering a question. Extending this insight to VLMs could help define how attention should behave in task-specific settings. By incorporating gaze collection protocols tailored to distinct tasks, we may be able to design models that not only align better with human attention but also adapt their internal focus in a task-aware manner.

In summary, our framework provides a strong foundation for integrating gaze as a training signal in egocentric VLMs. Future work can build on this by exploring task-specific attention modeling, leveraging insights from both vision science and cognitive psychology, and investigating how different forms of gaze data interact with visual representation choices across diverse downstream tasks.

