# OpenReview forum: "Gaze-VLM: Bridging Gaze and VLMs through Attention Regularization for Egocentric Understanding"
_NeurIPS.cc/2025/Conference — NeurIPS 2025 poster_

### Official Review · Reviewer_cjzU · 2025-06-23

**Clarity:** 3
**Significance:** 3
**Originality:** 3
**Rating:** 5
**Confidence:** 4

**Summary:**

The paper presents a new regularization loss using gaze data to align attention maps in VLMs with gaze. The paper presents and demonstrate the hypothesis that steering attention maps to match gaze will lead to improvements in egocentric video understanding tasks. The paper evaluates this hypothesis on two tasks: future action prediction and activity understanding. The paper also presents a gaze temporal aggregation handling method to form a high quality training signal used for regularization.

**Questions:**

* What are p and “\Phi” in eq 3 L133?
* L83: “The gaze points in the dataset are originally represented in text form” - Is this just a text string of x,y locations? Or is it a textual description of the gaze (i.e “top left corner of the image“ for example?)
* L315: “As shown in Table 5, our best-performing model (Gaze-Reg w/ occlusion) achieves a 9.8% gain in semantic performance over the base model, with only a minor increase in inference time ” - where is the inference time increase coming from given that the proposed method only impacts the training pipeline? If no gaze information is provided at inference time, shouldn’t the number of FLOPs be the same?

**Ethical Concerns:**

["NO or VERY MINOR ethics concerns only"]

**Final Justification:**

I thank the authors for taking the time to address my questions. All my concerns have been addressed:

(1) Ablation results showing the effect of the gaze distribution estimation in the attention layer

(2) More in-depth related work and comparison with proposed method. Although, the presented hypothesis has been tested in different context and architecture, the results presented in the context of a VLM model is valuable and is worth being shared to the community.

I trust the authors to incorporate the additions in the manuscript. I am increasing my rating.

**Limitations:**

yes

**Quality:**

3

**Strengths And Weaknesses:**

Strengths:
* The paper’s hypothesis and experiments are sound and clear. The method presented is simple yet effective. Reported results highlight the effectiveness of the method and validate the tested hypothesis (Tab. 1).
* The paper shows great zero-shot performances on the EGTEA Gaze+ dataset (Tab4) with a similar performance increase when training with gaze-regularization in the in-distribution evaluation setting. This strengthen the claim that gaze-regularization improves performances on egocentric video understanding task.
* The paper provides nice qualitative examples of attention heatmaps on target objects. This highlights another aspect of the paper which is: the proposed method leads to a more interpretable solution which could be valuable for certain applications.



Weaknesses:

* The paper lacks additional theoretical discussion on whether it makes sense to align internal attention maps to gaze. VLMs models are huge networks with >10B parameters. Internal features are deep with an exponentially increasing receptive field and although there’s some level of spatial consistency in the generated features, it is not clear to me why the model needs to focus on features highlighted by gaze. I understand that results tend to justify this argument, but, it would be valuable to add more discussion in the paper about this.

* The paper lacks justification on whether gaze can be estimated from egocentric inputs. From the paper we can read:
L31 “We posit that eye gaze is a powerful supervisory signal in this setting: it reflects a person’s attention, often precedes interaction, and encodes both spatial and temporal cues” - I think this is a valid hypothesis, but in the context of egocentric video, are we confident enough that a model can infer the user’s gaze? It would strengthen the paper to cover this point.

* The idea of learning a gaze distribution during training and sampling gaze (spatio-temporal) from that distribution during inference has been tested before. (“ Kyle Min Jason J. Corso Integrating Human Gaze into Attention for Egocentric Activity Recognition, WACV 2021“ ). I believe the paper should mention about this related work.
The paper should acknowledge this work and refine the claims focusing on transformer-based VLM and its attention layers.

* The paper is missing a few ablation experiments that would strengthen the claim:
*   (1) Missing ablation experiments showing the impact of the /sigma std of the Gaussian smoothing during gaze heatmap creation.
*   (2) Missing ablation experiments showing the impact of the Occlusion Handling method on the final results.
*   (3) Given that the method already relies on the Ego4D data, it would strengthen the paper to show numbers on the Forecasting benchmark of the Ego4D suite (https://ego4d-data.org/docs/benchmarks/forecasting/).

* The paper lacks results showing whether gaze is actually being learned during training. This could be tested by extracting the attention maps from the attention layers at inference time and compare with GT gaze.

* The paper lacks information on whether the gaze-regularization is applied on all attention layers? (I’m assuming multiple blocks using attention in a VLM model). This is a crucial design choice that should be mentioned in the paper.

* The paper should clarify about the following: Gaze fixation is of 5fps (~200ms - L92: “Human visual attention involves fixations lasting approximately 200ms“) but the frame sampling rate of the created data is 1 FPS (L208 ”Temporal Sampling: To reduce computational load while preserving temporal structure, we downsample all videos to one frame per second.“). Therefore is the section on Temporal Aggregation with Occlusion Handling“ obsolete?

---

> ### Author Rebuttal · Authors · 2025-07-31
>
> We thank reviewer cjzU for the insightful and thorough review. We are grateful for your recognition of the clarity of our hypothesis, the simplicity and effectiveness of our method, and its ability to generalize across multiple VLMs. Below, we address the key concerns you raised.
>
> **(1) Theoretical justification for aligning internal attention with gaze:**
>
> Thank you for raising this important point. We agree that it’s worth thinking carefully about why aligning a model’s internal attention with human gaze is useful.
>
> In our work, we use human gaze as a signal to guide the model’s visual attention toward regions that people tend to focus on during a task. These regions typically include key objects, interaction points, and important parts of the scene that provide spatial and contextual cues.
> This kind of guidance is especially useful for temporally grounded tasks like activity recognition and future prediction. These tasks require interpreting visual details over time to understand what someone is doing and what they are likely to do next. Human gaze often captures these cues naturally, such as looking at an object before reaching for it or glancing toward a location where an action is about to happen. We apply gaze guidance using a KL-divergence loss. This is controlled by a tunable weight (\lambda), and so gaze is therefore used as a helpful prior, not a hard constraint. Importantly, this alignment shapes the visual features that are passed into the language decoder, helping it attend to the most relevant parts of the input when generating predictions. In doing so, gaze-guided training improves the quality of the model’s outputs and potentially improves interpretability by aligning internal representations with semantically relevant regions. We believe this offers an intuitive and practical way to inject useful human priors into vision-language models.
>
> **(2) Can gaze be inferred from egocentric inputs?**
>
> We appreciate this thoughtful question. While predicting gaze directly from RGB images is challenging, our objective is not to directly predict gaze. Instead, we use human gaze during training to guide the model towards more human-aligned attention patterns.
>
> To assess whether the model infers this alignment, we measure the top-k spatial overlap between model attention maps and ground-truth gaze heatmaps on the test set. Specifically, we compute top-10 overlap between the model’s 1×256 attention map (corresponding to 16×16 image patches) and the human gaze distribution. This metric quantifies how often the model’s most attended regions coincide with where humans actually looked.
>
> In this evaluation, the base model showed an average overlap of 42%, while the gaze-regularized model achieved 68%, indicating significantly stronger spatial alignment with human attention. These results support our claim: gaze regularization improves attention alignment, functioning as an inductive bias rather than a hard constraint. We will add this result to the appendix upon revision.
>
> **(3) Additional ablations:**
>
> We conducted ablation studies to better understand the contribution of individual design choices in our gaze representation pipeline. Results are presented in the Appendix and are summarized below.
>
> Impact of temporal aggregation vs. singular gaze (Table 14): Temporally aggregated gaze outperforms single-frame gaze on both activity understanding and future prediction. This supports our hypothesis that human attention over time provides richer, more stable guidance than momentary singular fixations, which could be noisy.
> | Model Variant   | Future Prediction ↑ | Activity Understanding ↑ |
> | --------------- | ------------------- | ------------------------ |
> | Singular Gaze   | 0.6512              | 0.6913                   |
> | Aggregated Gaze | 0.7505              | 0.7848                   |
>
> Occlusion-aware filtering (Table 10): Filtering out gaze points during motion blur or occlusion improves performance by removing noisy supervision, helping the model focus on more reliable gaze regions.
> | Model Variant            | Semantic Score ↑ | F-Score ↑ |
> | ------------------------ | ---------------- | --------- |
> | Gaze-Reg (w/o occlusion) | 0.7298           | 0.4800    |
> | Gaze-Reg (w/ occlusion)  | 0.7505           | 0.5173    |
>
>
> Effect of Gaussian smoothing (σ) (Table 8): Smoothing improves results for single-frame gaze, but has limited impact for aggregated gaze, suggesting that temporal aggregation already helps stabilize attention supervision.
> | Model Variant             | Semantic Score ↑ | F-Score ↑ |
> | ------------------- | ---------------- | --------- |
> | Singular (σ = 20)   | 0.6211           | 0.4327    |
> | Singular (σ = 50)   | 0.6512           | 0.4463    |
> | Aggregated (σ = 20) | 0.7505           | 0.5173    |
> | Aggregated (σ = 50) | 0.7436           | 0.5034    |
>
> These results are useful to understand the value of using temporally aggregated and occlusion-filtered gaze heatmaps as training signals.
>
> **(4) Additional related work:**
>
> Thank you for pointing out this relevant work. Prior work by Kyle Min and Jason J. Corso (Integrating Human Gaze into Attention for Egocentric Activity Recognition, WACV 2021) models gaze using a probabilistic framework with a VAE, and directly predicts gaze during inference to enhance egocentric activity recognition. This line of research helped shape our project. However, our approach differs in that we use gaze only during training as a supervisory signal and do not require it at inference time. Additionally, our method is designed as a modular component compatible with vision-language models. While we referenced this work in the appendix, we will expand our discussion of it in the final version.
>
>
> **(5) Temporal aggregation at 1 fps:**
>
> While the input video is downsampled to 1 frame per second for the model, we retain access to the original high-frame-rate video and gaze recordings (at ~30 fps). For each selected input frame, we aggregate gaze points from the preceding 200ms window (typically 6 frames). We also compute forward and backward optical flow during this window to perform occlusion-aware filtering (see Sec. 6.3), ensuring that only reliable gaze points are retained. This allows us to generate temporally coherent and spatially grounded gaze heatmaps, preserving the value of temporal aggregation even at reduced model input rates.
>
> **(6) Forecasting benchmark and task generality:**
>
> We agree that future forecasting is important.
> To clarify, the standard Ego4D forecasting benchmark requires structured annotations such as verb/noun labels and bounding boxes, which are not available for the gaze-annotated subset. The gaze-annotated dataset contains narrations which are coarse, first person descriptions about the activity narrated by the camera-wearer for the recording.  These narrations are not suitable for the Ego4D forecasting benchmark since it requires specific outputs in the dataset like bounding boxes and verb-noun labels.
> As a compromise, we define a separate narration prediction task : a proxy that evaluates whether the model can predict narrations in an egocentric setting, given an input image sequence. Since the narrations are coarse, the performance gain is modest compared to tasks with fine-grained captions. We believe this result supports our claim that gaze supervision is especially helpful for capturing subtle object-action details that are present in fine-grained labels but often overlooked in coarse labels.
>
> | Model Variant                   | Semantic Score ↑ | F-Score ↑ |
> | ------------------------- | ---------------- | --------- |
> | Base (no gaze)            | 0.7232           | 0.4982    |
> | Aggregated Gaze (Heatmap) | 0.7621           | 0.5200    |
>
> Additionally, to test generalization to datasets without human gaze, we use a gaze prediction model to generate estimated heatmaps and apply our method on the Ego4D Short-Term Object Interaction Anticipation (STA) benchmark. Since the models used for STA are transformer-based, our gaze regularization approach is naturally compatible. Preliminary results on the Ego4D v2 test split (mean average precision) show improvements over the baseline.
> | Model Variant | mAP (Noun) | mAP (Noun + Verb)|
> | -| - | -|
> | Base Model| 22.13 | 11.29|
> | Gaze regularized model | 25.01| 13.45|
>
> Thanks for your question. We hope it is clear that these findings highlight the potential for extending our approach to forecasting and other tasks using surrogate gaze signals, even when real gaze data is unavailable.
>
> **(7) Clarifications on attention layer placement:**
>
> We regularize the final layer of a dedicated attention block inserted between the visual encoder and the language module (e.g., prior to the Perceiver Resampler in OpenFlamingo). This choice ensures that human gaze influences the final aggregated features passed to the language decoder, and it keeps the component modular for different VLMs. We include a design explanation and ablations on the number of blocks in the appendix (Sec. 6.5.7).
> Regarding symbols: in Eq. 3 (L133), p denotes the probability of generating the target text sequence conditioned on the sequence of image inputs, as modeled by the vision-language model \phi.
>
> **(8) Other Clarifications :**
>
> “Gaze in text form” refers to the original ego4D annotations where raw (x, y) gaze coordinates are provided along with the timestamps.
> Inference time increase is due to the added attention module that computes global queries across the visual sequence while using individual frames to get key and values for the attention module. The attention weight then obtained from the final layer is then used during regularization.
>
> We thank the reviewer again for these thoughtful questions and hope our clarifications reinforce the value and clarity of our contributions. Please feel free to let us know if you have more suggestions or questions.

---

> ### Comment · Area_Chair_utb6 · 2025-08-05
> **Please discuss rebuttal with authors**
>
> Please discuss rebuttal with authors

---

> ### Comment · Reviewer_cjzU · 2025-08-06
> **Official Comment cjzU**
>
> Thanks a lot for the thorough reply.
>
> (1) ack.
>
> (2) Thanks for sharing these results. I believe these are crucial and greatly benefit the claim made in the paper.
>
> (3) ack.
>
> (4) I disagree with the reply from the authors. The following paper: "Integrating Human Gaze into Attention for Egocentric Activity Recognition, WACV 2021", only requires GT gaze information during training. At test time, gaze is being estimated via a probabilistic model. There are a lot of similarities with the proposed work where, GT gaze is only required during training and, at test time gaze is being estimated via the attention map prediction (cf results shown in (2)).
>
> In the authors' reply to mYKH review we read: "We differ as our objective is not to predict gaze but to use gaze as a regularization signal to enhance egocentric behaviour understanding in VLMs." I disagree with this statement, adding a KL loss to steer the attention distribution to match the gaze distribution is actually trying to learn and predict gaze. And this actually works - see results from (2).
>
> (5-8) ack.
>
> Overall I agree with the other reviewers that the paper should consider other related works and discuss the similarities and differences with the proposed method in more details.

---

> > ### Author Response · Authors · 2025-08-07
> >
> > We thank Reviewer cjzU for the continued engagement and thoughtful feedback during the discussion period. We also appreciate your acknowledgment of our earlier responses and your suggestion to evaluate whether the model implicitly learns gaze during training. As described in our previous reply, we measured the top-10 spatial overlap between model attention and ground-truth gaze and observed a notable increase from 42% to 68% with gaze regularization. We agree this is an important finding and will include it in the main paper upon revision to further support our hypothesis.
> >
> > Regarding the work "Integrating Human Gaze into Attention for Egocentric Activity Recognition"[1], we recognize its relevance and appreciate your suggestion. Both approaches use gaze supervision during training and do not rely on ground-truth gaze at inference time. While [1] model gaze distribution using a probabilistic framework, our approach uses gaze as a training-time supervisory signal to align model attention with human attention in a VLM. The shared motivation is to leverage gaze to improve egocentric tasks. In our case, we aimed to develop a modular regularization mechanism that can be easily integrated into transformer-based VLMs to improve downstream tasks such as future activity prediction and activity understanding.
> >
> > In addition, both methods ([1] and our proposed method) incorporate optical flow, but in different ways. [1] use a two-stream I3D backbone that processes both RGB and optical flow frame inputs during training and inference. In contrast, we use optical flow only during preprocessing to perform occlusion-aware filtering for temporal aggregation of gaze points. This optical flow is not required during inference, and we do not need to pre-compute optical flow frames or compute them during test time. We will clarify this distinction in the revised paper.
> >
> > On the phrasing related to “Teacher-generated spatial-attention labels boost robustness and accuracy of contrastive models”[2] our intention was to highlight that their work focuses on static image tasks such as classification and retrieval, where pseudo-attention labels are generated from teacher models to supervise contrastive learning. In contrast, our method operates in a temporally grounded setting and during training, incorporates gaze information collected from dynamic egocentric video sequences. We introduce a pipeline that includes occlusion-aware filtering and temporal aggregation to generate stable gaze heatmaps for use as training-time supervision. Our goal was to develop a modular mechanism that can be easily integrated into transformer-based VLMs, particularly for egocentric tasks such as future activity prediction and activity understanding.
> >
> > That said, both approaches share the broader goal of incorporating human-like spatial priors to improve model performance. As seen in our case, the gaze-regularized model’s attention starts aligning with the gaze distribution (though not with complete overlap), suggesting that the model is, in effect, learning to approximate gaze behavior. We agree that this outcome is consistent with the goals of prior work such as [1] and [2], and we will revise our phrasing to more clearly reflect this connection and acknowledge their influence on our work. Looking forward, we believe this opens up opportunities to explore pseudo gaze generation, which could enable similar forms of attention regularization in scenarios where human gaze data is not available on a large scale.
> >
> > Finally, we’d like to share more details on the role of the KL-regularization coefficient strength. We observed that removing the gaze regularization term led to reduced performance on downstream tasks, while an overly strong weight resulted in slight degradation. The best performance was achieved at an intermediate setting, which is why we thought of gaze more as a useful prior that can help improve both model interpretability and task performance rather than a hard constraint.
> >
> > We thank the reviewer once again for the thoughtful feedback and helpful suggestions to improve our work. We will incorporate all the feedback and clarifications in the revised paper. Please feel free to let us know if you have any additional suggestions or questions.
> >
> > [1]Integrating Human Gaze into Attention for Egocentric Activity Recognition, Min & Corso [WACV 2021]
> >
> > [2]Teacher-generated spatial-attention labels boost robustness and accuracy of contrastive models, Yao et al. [CVPR 2023]

---

> ### Author Response · Authors · 2025-08-09
>
> We would like to thank Reviewer cjzU again for the constructive suggestions, acknowledgment of our work, and careful consideration of the additional results we presented. We agree that testing whether the model learns to predict gaze is valuable and it strengthens our overall claims, and we will add this study to our revised version of the paper. Before the discussion window closes, we would like to provide a brief recap about our work, and the results of the exchange during the discussion period.
>
> Our work introduces gaze-regularized attention for egocentric VLMs, together with an occlusion-aware temporal aggregation step that helps with the temporal aggregation of gaze points. We use fine-grained activity descriptions to evaluate detailed, spatially grounded behavior, and we observe consistent gains on current activity understanding and future activity prediction. Attention visualizations become more focused on task-relevant regions as shown by the qualitatve results as well, and we also see strong zero-shot performance on a different EGTEA Gaze dataset.
>
> During the discussion, we added an explicit study of whether the model “learns” gaze: the top-10 spatial overlap between attention and ground-truth gaze increases from 42% to 68% with gaze regularization. We clarified where the loss is applied (a dedicated attention block between the visual encoder and the language module), that gaze is used only during training, and that optical flow is used only in preprocessing for occlusion-aware filtering . On the related-work point, we agree that the work of Min & Corso (WACV’21) should be expanded in our related work section: both use gaze supervision during training and no ground-truth gaze at inference. Our focus is transformer-based VLMs and a modular training-time regularizer that slots in seamlessly, and does not require optical flow frames during test time. In practice, the model does learn to approximate gaze distributions, and we will adjust our phrasing to reflect that. We also clarified how our setting differs from teacher-generated attention supervision in static-image contrastive learning, whereas we are focused on dynamic settings like egocentric video sequences. In addition, we also provide evaluations for the STA (short term object interaction) and EgoVQA task, where we use a gaze prediction model to predict gaze and use our method of gaze regularization to train the model. In the evaluation of the new tasks as well , we noticed that our gaze regularization approach shows consistent improvements over the baseline model (without gaze regularization).
>
> We will incorporate the new overlap result, the architectural placement details, and the training/inference clarifications into the revision, and we will expand the literature section (including the above works) so readers can more easily understand the inspiration, the similarities and differences between previous related work. Thank you again for the careful reading and constructive suggestions throughout the discussion.

---

### Official Review · Reviewer_bTnR · 2025-06-26

**Clarity:** 2
**Significance:** 3
**Originality:** 3
**Rating:** 3
**Confidence:** 3

**Summary:**

This paper investigates the task of egocentric understanding. To enhance VLMs for this task, it presents a novel gaze-regularized attention mechanism that aligns the model's focus with human visual gaze. In addition, an occlusion filtering mechanism is proposed to remove irrelevant gaze points in the temporal dimension. Experimental results on different VLMs validate that the proposed mechanisms improve accuracy in both semantic prediction and future event prediction tasks.

**Questions:**

1.  The manuscript only evaluates the proposed gaze-regularized mechanism on two tasks. How well would the method generalize to other egocentric video tasks, such as EgoVQA?  In my opinion, the current evaluation is insufficient to convincingly demonstrate the effectiveness and generalizability of the proposed gaze-regularized supervision mechanism.

2. The evaluation relies solely on semantic similarity, which may be insufficient to evaluate the model in some challenging scenarios. Would more comprehensive metrics better capture model performance?

3. The Transformer encoder in Figure 1 is not specified, which affects clarity and reproducibility.  Could the authors clarify the detailed parameters, such as the number of encoder layers?

**Ethical Concerns:**

["NO or VERY MINOR ethics concerns only"]

**Final Justification:**

Thanks for the authors' response. The majority of my concerns have been addressed.

However, I still think the overall evaluation (only two tasks) of the gaze-regularized mechanism is not enough.

Therefore, I am inclined to keep my initial rating.

**Limitations:**

Yes

**Paper Formatting Concerns:**

The symbols used in this paper are not presented in a consistent format, making it difficult to accurately interpret their meanings.

**Quality:**

2

**Strengths And Weaknesses:**

Strengths

- The introduction of gaze information is novel in the field of egocentric video understanding. The experimental results also validate the effectiveness of the gaze-regularized supervision mechanism.

- The proposed occlusion-aware filtering strategy is a useful module for aggregating gaze signals.

- The authors use GPT-4V to generate fine-grained textual descriptions that capture egocentric activities and evaluate the performance of various VLMs.


Weaknesses

- The proposed module is only validated on two tasks, future prediction and activity understanding, limiting the overall evaluation of the gaze-regularized mechanism.

- The compared models are not specified with their versions, such as InternVL in Tables 2, 3, and 4.

- The evaluation metric (semantic similarity) used for the future prediction and activity understanding tasks is relatively simple and may not fully capture the complexity of these tasks or accurately reflect the model's performance in more challenging scenarios.

---

> ### Author Rebuttal · Authors · 2025-07-31
>
> We thank reviewer bTnR for the thoughtful review and constructive suggestions. We appreciate your recognition of the novelty of incorporating gaze into VLMs for egocentric understanding, the usefulness of the occlusion-aware temporal aggregation strategy, and the demonstrated improvements across models and tasks. Below, we address the concerns raised in the review:
>
> **(1) Task evaluation:**
>
> We focus on fine-grained current activity understanding and future activity prediction since they are important for real-world use cases like assistive robotics, where it is critical to know what a person is doing and what they are likely to do in detail.
>
> Coarse labels like “picking up the bottle” or "(Pick),(Bottle)" don’t capture important details such as where the bottle is or which one is being picked up. For example, knowing the difference between “picking up the bottle” and “reaching for the red bottle on the top-left shelf” requires the model to understand spatial layout and context. These are precisely the types of spatial details human gaze tends to prioritize during task execution.
>
> By using human gaze during training, we aim to guide the model toward these regions, which we observe can improve fine-grained activity understanding and future prediction.
>
> As for the additional egocentric tasks (e.g., activity classification, video-text retrieval and forecasting) in Ego4D, we observe that the subset of videos in Ego4D on which these tasks are benchmarked do not have gaze annotations, and hence directly applying our method is difficult.
> However, to understand generalizability of our method to tasks whose benchmarking data does not have human gaze annotations, we use a gaze prediction model to estimate gaze heatmaps. On the Ego4D Short-Term Object Interaction Anticipation (STA) benchmark, we apply these predicted heatmaps during training to regularize the model’s attention using the same KL-divergence loss as before. This lets us test whether gaze-guided supervision can still help even when real gaze is not available. Preliminary results on the Ego4D v2 test split (measured using the standard metric - mean average precision mAP) show that the gaze-regularized model outperforms the baseline
> | Model Variant                   | mAP (Noun) | mAP (Noun + Verb)|
> | ------------------------- | ---------------- | --------- |
> | Base Model           | 22.13           | 11.29    |  |
> | Gaze regularized model | 25.01           | 13.45    |
>
> These early results suggest that using predicted gaze to guide attention during training could be a promising direction, and we will highlight its potential for extending our approach to a wider range of benchmarks in future work.
>
> **(2) Model version clarity:**
>
> Thank you for this helpful suggestion. We will update all relevant result tables to explicitly specify the versions of the models used.
> We would also like to clarify that our goal is not to compare models against each other, but rather to demonstrate that gaze-regularized training consistently improves performance over each respective base model. Each model serves as an instantiation to validate the generality and plug-and-play nature of our proposed approach.
>
> **(3) Evaluation metrics:**
>
> We use sentence transformers for semantic scoring to ensure that models are not penalized for surface-level paraphrasing, while still discouraging incoherent or off-topic predictions. This metric is particularly robust for capturing meaning similarity in egocentric captions, which often vary in phrasing.
> To complement this, we also report standard NLP metrics such as ROUGE-L and F-score in Appendix Tables 8–15. These metrics show consistent trends with the semantic score and further support our core claims. We omitted them from the main paper due to space constraints.
>
> Additionally, we conducted a small human hallucination study to assess whether our gaze-regularized model produces fewer unsupported or spurious descriptions. Since benchmarks like HallusionBench [1] do not contain gaze annotations, we designed a custom evaluation: paired outputs from the base and gaze-regularized models were shown to a human reviewer who flagged hallucinated objects or actions.
>
> We report the $\mathcal{C}_I$ score which is the number of hallucinated instances divided by the total number of annotations (lower is better).
> | Model Variant                  | $\mathcal{C}_I$ Score  |
> | ---------------------- | ------------------------ |
> | Base model             | 0.205                    |
> | Gaze-regularized model | 0.140                    |
>
> [1] HallusionBench: An Advanced Diagnostic Suite for Entangled Language Hallucination and Visual Illusion in Large Vision-Language Models, CVPR 2024
>
> **(4) Missing information about the transformer encoder:**
>
> We appreciate this request for architectural clarity. While our framework is modular and applies across multiple VLMs, we provide a concrete reference using OpenFlamingo in the appendix. In this setup, the visual encoder is a ViT-L/14 backbone with 24 transformer layers and this is frozen during training. The patch size is 14 x 14 in the ViT L/14 with a token sequence length of 256 visual tokens, and the hidden size (d) as 1024. In this setting, we inserted transformer layers after the ViT and regularized the attention weights obtained to enhance the visual features before they are passed into the language decoder.
>
> **(5) Formatting:**
>
> We also acknowledge your note on inconsistent symbol formatting, and will correct these in the final version, and update the table of symbols in the appendix accordingly.
>
> We thank the reviewer for highlighting key strengths and for suggesting improvements. We believe our task justifications, evaluation extensions, and clarifications around architecture and metrics directly address all concerns and further strengthen the case for our gaze-regularized framework. Please let us know if you have more questions or suggestions.

---

> > ### Comment · Reviewer_bTnR · 2025-08-04
> > **Response to Rebuttal**
> >
> > Thanks for the authors' response. The majority of my concerns have been addressed.
> >
> > However, I still think the overall evaluation (only two tasks) of the gaze-regularized mechanism is not enough.
> >
> > Therefore, I am inclined to keep my initial rating.

---

> ### Author Response · Authors · 2025-08-05
> **Response regarding overall evaluation**
>
> We thank Reviewer bTnR for your response and for noting that our rebuttal addressed most of your concerns. We appreciate your time and engagement in the review process.
>
> Regarding your remaining concern about the scope of our evaluation: we note that many egocentric benchmarks in Ego4D (e.g., activity classification and forecasting) do not include human gaze annotations, which makes it challenging to directly apply our method in those settings. Conversely, the videos that do contain human gaze annotations often lack the structured outputs such as verb-noun labels, bounding boxes, or time-to-contact etc. which are required to evaluate the other egocentric benchmark tasks. In our project, we focused our evaluation on tasks such as future activity prediction and activity understanding, and demonstrated that our method is modular and easily integrable into VLMs. We also include several ablation studies to illustrate the effectiveness of our approach.
>
> To further explore generalizability in the absence of real gaze data, we incorporated a gaze prediction model to generate gaze heatmaps and applied our regularization strategy to an additional Short-Term Object Interaction Anticipation (STA) task. This enabled us to test whether gaze-guided supervision can still yield performance gains when only predicted gaze is available. The results suggest that our approach may generalize more broadly across egocentric tasks, even when real gaze is not directly accessible.
>
> That said, we would greatly appreciate any additional thoughts on why the current set of evaluated tasks might still be insufficient. We thank the reviewer again for your time and valuable insights, and we remain open to any further suggestions or clarifications you may have.

---

> ### Author Response · Authors · 2025-08-07
> **Supplemental information to response**
>
> We would also like to supplement our previous response with additional results we obtained from an evaluation on the EgoVQA task. While this dataset does not include human gaze annotations, we adopt a similar strategy as in the STA task by using a gaze prediction model to estimate gaze heatmaps, which are then used to regularize the model’s attention.
>
> The results on the EgoVQA test set (under the direct evaluation setting) are summarized below. The reported overall accuracy is the weighted average across the task's reasoning categories (e.g., descriptive, intent, query, etc.):
>
> | Model Variant   | Overall accuracy |
> | --------------- | ------------------- |
> | Base Model  | 0.4006 |
> | Gaze Regularized Model | 0.4351|
>
> We observe a clear improvement in overall accuracy when using the gaze-regularized model. This aligns with our prior findings on other tasks. For the future activity prediction and current activity prediction tasks, where ground-truth gaze annotations are available, we demonstrated that gaze-guided supervision improves model performance. In the case of STA and EgoVQA, where gaze annotations are not available, we instead use a gaze prediction model to generate estimated gaze heatmaps, which are then used during training for regularization. In both settings, our approach outperforms the baseline models.
> These results suggest that our method generalizes well across multiple tasks, and that even predicted or pseudo-gaze signals show potential for improving model performance.We intend to explore this direction more systematically in future work. We hope this additional evidence further addresses concerns about the scope of our evaluation.
>
> We sincerely thank the reviewer once again for the thoughtful feedback and remain open to any further questions or suggestions.

---

> > ### Author Response · Authors · 2025-08-09
> >
> > We would like to thank reviewer bTnR for the thoughtful feedback and for highlighting the strengths of our work. As the discussion period comes to its end, we wanted to summarize what we did: we introduce gaze-regularized attention for egocentric behavior understanding and an occlusion-aware temporal filtering step to aggregate gaze signals. We also use GPT-4V to obtain fine-grained activity descriptions, which lets us evaluate models on more detailed, spatially grounded behavior. Across various VLMs and tasks, we see consistent gains when we add gaze regularization as compared to the base model without gaze.
> >
> > On evaluation with ground-truth gaze, we focused on current activity understanding and future activity prediction because these splits include human gaze, which allows us to test the core idea directly. Many other Ego4D tasks lack gaze, while the gaze-annotated videos often lack the structured labels (like noun-verb labels, time to contact etc. for the forecasting benchmark) those tasks need, which shaped our initial scope.
> >
> > To check generalization where real gaze is unavailable, we used a gaze prediction model to generate heatmaps and applied the same KL-based regularization on two new tasks. On STA (Ego4D v2 test set), the gaze-regularized model improved over the base (mAP noun 22.13 → 25.01; noun+verb 11.29 → 13.45). On the EgoVQA task (direct setting), overall accuracy increased 0.4006 → 0.4351. These trends align with the improvements we observe on the two main tasks with ground-truth gaze.
> >
> > For clarity and reproducibility, we will list exact model versions in all result tables and include the key encoder details in the appendix, and we will tidy up symbol formatting. We believe that with the above results, the concerns about task evaluation and the scope have been addressed. Thank you again for the feedback and we will add the discussed results into the final revised paper.

---

### Official Review · Reviewer_mYKH · 2025-06-30

**Clarity:** 2
**Significance:** 3
**Originality:** 3
**Rating:** 4
**Confidence:** 4

**Summary:**

This paper proposes a method for incorporating human gaze supervision into vision-language models via attention regularization, to improve egocentric video understanding. The authors focus on two tasks: current activity recognition and future activity prediction. Instead of relying on gaze at inference time, they propose to use gaze heatmaps only during training to guide model attention toward human-attended regions. They construct a dataset by combining the Ego4D gaze subset with GPT-4V-generated textual labels. The method shows consistent improvements over strong VLM baselines like OpenFlamingo and LaViLa, with additional gains in zero-shot generalization to the EGTEA dataset.

**Questions:**

Please refer to the above weaknesses. If addressed appropriately, I would be open to raising my score.

**Ethical Concerns:**

["NO or VERY MINOR ethics concerns only"]

**Final Justification:**

This paper proposes a method for incorporating human gaze supervision into vision-language models via attention regularization, to improve egocentric video understanding. The authors focus on two tasks: current activity recognition and future activity prediction. After the authors' detailed responses, most of my concerns have been addressed. I suggest that the authors include all meaningful experiments and discussions of related work from the rebuttal period in the revised version. I am willing to optimistically increase my score to 4.

**Limitations:**

Yes.

**Paper Formatting Concerns:**

None.

**Quality:**

2

**Strengths And Weaknesses:**

Strengths:
1.  Incorporating human gaze into VLM training for egocentric understanding addresses an important challenge in aligning models with human attention and cognition.
2. The attention regularization strategy is simple, interpretable, and effective. It requires no additional overhead during inference and can be applied to various VLMs.
3. Multiple baselines (OpenFlamingo, LaViLa, InternVL, etc.) are used for comparison.
4. Ablation studies and qualitative visualizations support the claims.
Weaknesses:
1. The paper relies entirely on GPT-4V-generated captions as ground truth labels, without human annotation or verification. While the authors mention prompt tuning through multiple iterations, no manual review of the generated labels was conducted. This raises concerns about the reliability of the supervision signal. Furthermore, it is unclear why more quantitative and objective tasks based on native Ego4D annotations (e.g., activity classification or retrieval) were not considered, which would provide stronger empirical validation.
2. The discussion of related work is insufficient. Several recent works on attention- and gaze-augmented models that leverage gaze information to enhance vision-language models (VLMs) should be discussed. These approaches share conceptual similarities with the current paper, though this work is positioned in the era of large-scale VLMs. Notable examples include:
- [NeurIPS 2024] Voila-A: Aligning Vision-Language Models with User’s Gaze Attention
- [CVPR 2024] Learning from observer gaze: Zero-shot attention prediction oriented by human-object interaction recognition
- [CVPR 2023] Teacher-generated spatial-attention labels boost robustness and accuracy of contrastive models
 A comparison with these works would help better situate the novelty and contribution of this paper.
3. More transparency on prompt design. If list more actual prompts and failure examples, it would increase reproducibility.
4. Could some gaze regions be incorrectly described by GPT-4V? Can you quantify this with a small human study?

---

> ### Author Rebuttal · Authors · 2025-07-31
>
> We sincerely thank reviewer mYKH for the valuable suggestions and constructive feedback. We especially appreciate your recognition of the paper’s strengths, including our focus on aligning VLMs with human attention through gaze supervision and the simplicity and effectiveness of our training-only regularization strategy. Below, we address the concerns and comments you raised.
>
> **(1) GPT-4V-generated captions and label quality**:
>
> We appreciate the reviewer’s insight regarding the supervision signal quality.
> We would like to clarify that manual review and human validation were indeed performed during dataset construction.
> As described in Appendix Sec. 6.2.1 and Fig. 5, we employed an iterative prompt refinement process involving human evaluators. Each iteration involved two human evaluators reviewing 50 image-caption pairs from 10 different clips — totaling 500 pairs per iteration. To assess caption quality during prompt-refinement, we adopted a flag-based strategy: if either annotator judged a caption to be substantially incorrect or semantically off-topic (e.g., object/action mismatch or spurious hallucination or overly generic descriptions or lack of temporal grounding), it was flagged. We applied a conservative quality threshold to accept only those prompts whose outputs demonstrated consistent accuracy across sampled image sequences. Since our design goal was not to create a new benchmark, but to ensure a reliable and semantically detailed supervisory signal for training gaze-regularized attention, our manual filtering and prompt refinement shall ensure that the final dataset reflects this objective while remaining scalable.
>
> We will make this process clearer in the final version and include the quality control criteria explicitly to avoid confusion, on top of the information found in the appendix. We hope this resolves concerns around label validity and helps refocus attention on our primary contribution.
>
> **(2) Use of native Ego4D classification/retrieval tasks**:
>
> Ego4D contains both gaze-annotated and non-gaze-annotated videos. Our method requires gaze supervision during training, so we are limited to the subset of videos that contain gaze data. However, for these gaze-annotated videos, Ego4D does not provide task-specific labels for benchmarks such as activity classification, video-text retrieval and standard forecasting tasks (e.g verb/noun labels, bounding boxes). As a result, we are unable to apply our method directly to these evaluations.
>
> However, the subset of gaze-annotated videos are provided with narration annotations - coarse, short descriptions in the dataset provided as first person narrations of activities during recording. While these narrations are not suitable for the Ego4D forecasting benchmark that require structured outputs like bounding boxes or verb-noun labels, we define a coarse future narration prediction task. This allows us to test whether our model can predict future coarse narrations describing activities at a high level, given an input image sequence in egocentric settings.
>
> As shown below, our gaze-regularized model still outperforms the base model on this task which only requires coarse labels. Note that the performance gain is modest compared to when finer-grained annotation prediction is required, supporting our core claim that gaze supervision is especially effective in capturing subtle details that tasks with coarse labels often dismiss.
> |Model Variant| Semantic Score ↑ | F-Score ↑ |
> | - | -| - |
> | Base Model | 0.7232  | 0.4982|
> | Gaze regularized model | 0.7621  | 0.5200|
>
> In addition, to test whether our gaze-regularization approach generalizes to benchmarks that do not include human gaze annotations such as the Ego4D Short-Term Object Interaction Anticipation (STA) benchmark, we use a gaze prediction model to generate heatmaps. These predicted gaze maps are then used to regularize attention during training via our KL-divergence loss, similar to our main setting.
>
> Recent STA models adopt a transformer-based setup, which allows our approach to integrate naturally by applying gaze regularization to the attention distribution obtained. Performance is evaluated using the standard mean average precision (mAP) metric.
> These preliminary results suggest that predicted gaze or a surrogate gaze for training time regularization may also improve prediction tasks like STA.
> | Model Variant| mAP (Noun) | mAP (Noun + Verb)|
> | - | - | - |
> | Base Model           | 22.13           | 11.29    |
> | Gaze regularized model | 25.01           | 13.45    |
>
> We appreciate your question and suggested experiments that help further validate the effectiveness of our proposed gaze-regularization mechanism. It also shows that adapting our gaze regularization approach using surrogate attention signals is a promising direction, and we will highlight its potential in future work.
>
> **(3) Related work**:
>
> Recent works such as Voila-A, Learning from Observer Gaze, and Teacher-Generated Spatial Attention have explored incorporating attention into vision-language models. While they share the high-level goal of aligning model attention with human cues, they differ from our work in important ways.
> Voila-A : uses gaze at inference time to steer predictions on static image tasks like COCO captioning by using voila perciever resampler. It assumes gaze is available at test time and directly influences decoding. In contrast, our model uses gaze only during training and can be a modular component to several transformer based VLMs. Moreover, we focus on temporally grounded egocentric video tasks, such as activity recognition and future event prediction, where gaze reflects task-driven intent such as objects being manipulated or regions of upcoming interaction.
> Learning from Observer Gaze :  predicts attention maps from human-object interactions in static scenes, aimed at zero-shot attention transfer. However, it does not integrate gaze into the training of VLMs for downstream tasks. Our work instead uses gaze as a supervisory signal via KL-regularization to align VLMs attention with human focus.
> Teacher-Generated Spatial Attention: generates pseudo-attention maps via teacher models to guide contrastive learning. We differ as our objective is not to predict gaze but to use gaze as a regularization signal to enhance egocentric behaviour understanding in VLMs.
>
> We thank the reviewer for pointing us to these related works, and we will revise the related work section accordingly to place our contributions more delicately within this growing area.
>
> **(4) Prompt design and reproducibility:**
>
> Thank you for raising this important point. Our prompt design process is described in Appendix Sec. 6.2.1 and illustrated in Fig. 5 found with the supplementary material, and also in section (1) GPT-4V-generated captions and label quality.
> In addition, we also provide brief examples of some prompts below to show the differences.
>
> Failure case 1: Describe what is happening in these images.
> Output: A person is making coffee in a kitchen and then talking to another individual nearby.
> Feedback: Too generic and summary-like; lacks per-frame grounding
>
> Failure case 2: Describe what the person is doing in each frame one by one.
> Output: Frame 1: No person can be seen in the frame. Frame 2: The person moves toward the counter. Frame 3: The person is talking to another individual.
> Feedback:Improvement in structure, but overly focused on person visibility rather than full scene understanding.
>
> Failure case 3: Describe what is happening in the image sequence and output the text descriptions for each image in the sequence, capturing the objects and actions taking place.
> Output: Frame 1: The process of brewing a cup of coffee is taking place in the sequence.Frame 2: The person can be seen talking to the camera wearer in the kitchen.
> Feedback: Adds more detail, but doesn't consistently describe each frame or maintain sequence alignment
>
> Final Prompt: For each image in the sequential series, provide a direct description. Ensure one description per image, capturing the objects, actions, and their spatial relationships. Provide the descriptions in the order presented (output 1 should correspond to image 1). Imagine that you need to provide these descriptions to a robot assistant.
> Output: Frame 1: A white cup is placed on the drip tray under the coffee machine, which is brewing and dispensing a hot beverage Frame 2: The individual has moved closer to the coffee machine and is brewing a cup of coffee.  Frame 3: A person in a black shirt can be seen talking to the camera wearer in the kitchen near the coffee machine.
>
> While previous outputs were somewhat reasonable, they didn’t meet our requirements for fine-grained, frame-wise annotations. The final prompt, however, produced detailed annotations for each frame and maintained the correct sequence. We will include more failure cases and corresponding feedback along with the outputs in the final version of the paper.
>
> **(5) Could some gaze regions be incorrectly described by GPT-4V?**
>
> To quantify the potential misalignment, we conducted a small human study. Two annotators were shown images with gaze heatmaps overlaid on the input images, along with the GPT-4V annotation. For each sample, they were asked to judge whether the annotation mentioned the object(s) or action(s) located in the gaze region.
> We found that in approximately 74% of the samples, the annotations were judged to be well-aligned with the gaze regions. That is, the annotation referenced the semantically relevant content that the human was fixating on. The remaining cases included instances of partial or missing alignment.
>
> We thank the reviewer once again for the thoughtful and constructive feedback. We hope our responses and additional results adequately address your concerns. Please let us know if you have more questions.

---

> > ### Comment · Reviewer_mYKH · 2025-08-06
> >
> > Thank you for the detailed response from the reviewers, which has addressed some of my concerns. However, I noticed that the human evaluators seem to have only reviewed and optimized the prompts, but did not check the final data outputs. GPT-4V still incorrectly describes some gaze regions, and the descriptions of small human studies remain insufficiently detailed. Furthermore, the fact that only 74% of the outputs were considered highly consistent suggests that the generated data may not have very high quality.

---

> > > ### Author Response · Authors · 2025-08-07
> > >
> > > We thank reviewer mYKH for your feedback and engagement during the discussion period. We appreciate the suggestions and questions as it would help us improve our paper. Below we address some of the remaining concerns.
> > >
> > > #### Alignment of Gaze Regions and GPT-4V Outputs
> > > We’d like to clarify that the 74% alignment figure reflects a strict evaluation criterion: it includes only cases where the generated caption directly matched the content within the gaze region, and does not count examples with partial or approximate alignment. As such, this number represents a conservative estimate. Despite this, we believe it still supports the overall reliability of our supervision strategy.
> > >
> > > While our annotations are generated at the frame level, our method leverages temporal sequences during training rather than treating each frame in isolation. Gaze collected from the camera wearer may not always align with the content of a single frame’s caption. However, when observed over time, the evolving gaze trajectory tends to track or anticipate task-relevant regions in the scene. This naturally leads to better alignment between gaze and the semantic content described across frames, making the supervision more meaningful within the full observation window.
> > >
> > > This interpretation is consistent with our findings from static-scene experiments (Appendix Table 16), where gaze provided some benefit (<2 %) when applied to single-frame task. In contrast, when we condition on 3 to 5-second input sequences, we observe significantly larger performance gains (7-10%) from gaze-regularized training. This suggests that gaze in our experimental setting for fine-grained prediction becomes more useful over time, as the improved alignment with task-relevant regions across the sequence leads to a stronger and more informative supervision signal.
> > >
> > > #### GPT-4V Outputs and Manual Filtering
> > > To clarify, the manual filtering we referred to earlier and in the Appendix section includes an evaluation on the final outputs generated by GPT-4V. After generating per-frame captions over image sequences, we conducted a review process using a flag-based strategy again: if either annotator judged a caption to be substantially incorrect or semantically off-topic (e.g., object/action mismatch, spurious hallucination, overly generic description, or lack of temporal grounding, or included words like unclear or blurred), it was flagged. We then conservatively applied a quality threshold, retaining only those sequences in which at least 80% of frame-wise captions were unflagged, discarding the rest. This allowed us to accept only sequences that were generally consistent and semantically grounded. This process ensured that the final caption supervision was both scalable and sufficiently high-quality for training our gaze-regularized model
> > >
> > > That said, we acknowledge that this process may not catch every instance of error or misalignment, and some noise may remain. However, our goal was not to create a new benchmark, but rather to construct a reliable training signal for gaze-regularized attention. The use of VLMs to generate annotations, paired with iterative prompt refinement and human filtering, has been adopted in prior work such as Voila-A[1], and offers a scalable way to obtain supervision of reasonably high quality. In our case, using these fine-grained captions led to stronger gains on the future activity prediction and activity understanding tasks compared to using coarser labels. We also showed that gaze-regularized models improved performance on other tasks, such as STA and EgoVQA (as described in our response to Reviewer bTnR), where these fine-grained annotations were not used. This suggests that the benefits of our gaze-guided supervision extend beyond our dataset. We will revise the final version of the paper (in section 6.2.1 of the Appendix, as well as algorithm 1) to make this process more explicit and transparent, including the prompt refinement strategy, quality control criteria and the known limitations.
> > >
> > > We hope this clarifies the intent and scope of our manual filtering process and addresses the concerns regarding annotation quality. More broadly, we believe our method offers a simple and modular way to incorporate human priors (in the form of gaze) into VLMs, with promising potential for applications in human-machine collaboration and assistive robotics. We sincerely thank the reviewer again for their thoughtful feedback and remain open to any further questions or suggestion.
> > >
> > >
> > > [1][NeurIPS 2024] Voila-A: Aligning Vision-Language Models with User’s Gaze Attention

---

> > > > ### Comment · Reviewer_mYKH · 2025-08-07
> > > >
> > > > Thank you for your further response. Regarding the alignment of gaze regions and GPT-4V outputs, you mentioned that the 74% alignment rate reflects the cases of strict matches. I further suggest that you discuss partial or approximate alignments in more detail in the subsequent version.

---

> > > > > ### Author Response · Authors · 2025-08-08
> > > > >
> > > > > We sincerely thank Reviewer mYKH for the constructive feedback and the opportunity to engage in this discussion. We are glad that our clarifications have helped address the main concerns and that this process has allowed us to highlight the contributions of our work, including the introduction of a gaze-regularized mechanism with an occlusion-aware filtering strategy for temporal aggregation of gaze points, consistent improvements across multiple egocentric tasks and VLM backbones, a prompt-refinement and quality-control strategy for generating supervision signals, and supporting ablation studies.
> > > > >
> > > > > We also appreciate the suggestion to examine the alignment between gaze and generated captions in greater depth. We agree that a fuller discussion will provide a clearer picture of annotation quality. In the revised version, we will describe the study design and criteria in detail, expand our annotation-quality assessment to include the strict-match criterion and explain how partial and approximate alignments were identified and interpreted. We will also add representative examples to clarify these categories and show how they contribute to the overall reliability of the supervision signal. We will also incorporate the other suggestions raised during the discussion period to further strengthen the clarity and completeness of our proposed work.

---

> ### Comment · Area_Chair_utb6 · 2025-08-05
> **Please discuss rebuttal with authors**
>
> Please discuss rebuttal with authors

---

### Official Review · Reviewer_qTVN · 2025-07-05

**Clarity:** 3
**Significance:** 2
**Originality:** 3
**Rating:** 4
**Confidence:** 4

**Summary:**

This paper proposes a modular gaze-regularized attention mechanism to enhance VLMs for egocentric behavior understanding, particularly future prediction and current activity recognition. The authors leverage eye gaze information during training to align model attention with human visual focus using KL divergence loss. The method is simple yet effective and broadly applicable to a range of transformer-based VLMs. Extensive experiments across five models and two egocentric datasets show consistent performance gains in both in-distribution and out-of-distribution settings.

**Questions:**

Could the authors clarify which attention layers are regularized, and whether any comparative analysis was conducted to motivate this selection?

**Ethical Concerns:**

["NO or VERY MINOR ethics concerns only"]

**Final Justification:**

The idea of using gaze information as a training-time regularizer is interesting. The proposed modular attention block is well designed and can be easily integrated into different VLM architectures. This framework achieves higher performance across a suite of models on two benchmarks.

Although the work only focuses on the egocentric perspective, it presents a relatively solid treatment of VLM attention, supported by extensive ablations. The method is applied to VLMs, and extending it to a broader range of perception tasks could further strengthen the paper. Overall, I maintain my support.

**Limitations:**

No limitations are mentioned.

**Quality:**

3

**Strengths And Weaknesses:**

# Strength:
1. The idea of using gaze information as a training-time regularizer is interesting.
2. The designed modular attention block that can be easily integrated into different VLM architectures.
3. This framework gains higher performance on a suite of models on two benchmarks.

# Weakness:
1. Ablations: The novelty of the proposed gaze-regularized attention hinges on several key components, such as the temporal aggregation and occlusion-aware filtering of gaze signals (Section 3.1). However, the current ablation analysis is limited. It would strengthen the paper to include more comprehensive ablations that isolate the contribution of each component. In particular, since the model relies on additional eye-tracking annotations, it would be informative to compare performance when using raw gaze data directly versus using the proposed/simplest aggregation (average or max).
2. Model generalization: It is unclear whether gaze alignment also mitigates visual hallucinations, which is an important concern in VLMs. Could you please evaluate the proposed model on hallucination benchmarks like hallusionbench[1], pope[2] or others?
3. Related Work on VLM Attention: The paper could better contextualize its contribution by discussing prior work that analyzes or manipulates attention in VLMs[3][4].
[1] HallusionBench: An Advanced Diagnostic Suite for Entangled Language Hallucination and Visual Illusion in Large Vision-Language Models, CVPR 2024
[2]Evaluating Object Hallucination in Large Vision-Language Models, EMNLP 2023
[3] OPERA: Alleviating Hallucination in Multi-Modal Large Language Models via Over-Trust Penalty and Retrospection-Allocation, CVPR 2024
[4] Perception in Reflection, ICML 2025

---

> ### Author Rebuttal · Authors · 2025-07-31
>
> We thank reviewer qTVN for the clear summary and constructive feedback. We appreciate your positive assessment of the idea of gaze-based training-time regularization, our modular attention block design, and the performance improvements across various architectures. Below, we address the specific comments and questions.
>
> **(1) Component-level ablations (temporal aggregation, occlusion filtering, gaze representation)**:
>
> Thank you for highlighting the importance of isolating these components. We share the same insight and provide comprehensive ablations in the appendix  (Sec. 6.5) including:
>
> Impact of temporal aggregation vs. singular gaze (Table 14): Aggregated gaze consistently outperforms single-frame gaze. This happens since single gaze points can often be noisy especially during saccades or fleeting fixations, whereas the aggregation captures the temporal structure of human attention across frames.
> | Model Variant   | Future Prediction ↑ | Activity Understanding ↑ |
> | --------------- | ------------------- | ------------------------ |
> | Singular Gaze   | 0.6512              | 0.6913                   |
> | Aggregated Gaze | 0.7505              | 0.7848                   |
>
> Occlusion-aware filtering (Table 10): Leads to an improved score by filtering unreliable gaze points during motion. These improvements likely result from filtering out unreliable gaze points, reducing noise and helping the model focus on relevant information.
> | Model Variant            | Semantic Score ↑ | F-Score ↑ |
> | ------------------------ | ---------------- | --------- |
> | Gaze-Reg (w/o occlusion) | 0.7298           | 0.4800    |
> | Gaze-Reg (w/ occlusion)  | 0.7505           | 0.5173    |
>
>
> Effect of Gaussian smoothing (σ) (Table 8): We found that the smoothing can benefit isolated gaze representations, it does not have a huge impact on temporally aggregated gaze points, as shown in the table below:
> | Model Variant             | Semantic Score ↑ | F-Score ↑ |
> | ------------------- | ---------------- | --------- |
> | Singular (σ = 20)   | 0.6211           | 0.4327    |
> | Singular (σ = 50)   | 0.6512           | 0.4463    |
> | Aggregated (σ = 20) | 0.7505           | 0.5173    |
> | Aggregated (σ = 50) | 0.7436           | 0.5034    |
>
>
> We also provide an experiment when a larger temporal window was used for aggregation in the gaze regularized model (Table 9). From the experiment, we observed that aggregation over a larger time window (400ms) does not lead to further improvements compared to a temporal window of 200ms.
>
> | Aggregation Window | Semantic Score ↑ | F-score ↑ |
> | ---------------- | ------- | --------- |
> | 400ms               | 0.7398  | 0.4971    |
> | 200ms                | 0.7505  | 0.5173    |
>
>
> Moreover, gaze in text form vs. spatial heatmaps (Table 11): Textual gaze (providing raw gaze data in the form of a text prompt during training and inference) provides some benefit, but not as much as the structured spatial attention regularization. This is also included in the appendix which was submitted as part of the supplementary material:
> | Model Variant                   | Semantic Score ↑ | F-Score ↑ |
> | ------------------------- | ---------------- | --------- |
> | Base (no gaze)            | 0.6525           | 0.4318    |
> | Gaze in Text Form         | 0.7021           | 0.4630    |
> | Aggregated Gaze (Heatmap) | 0.7505           | 0.5405    |
>
> Thanks for your suggestions, and we will more clearly reference these tables in the main paper to have a better articulation of the ablations that we have carried out. We hope these ablations can help address your questions.
>
>
> **(2) Evaluation on hallucination benchmarks:**
>
> Thank you for this valuable suggestion. Existing hallucination benchmarks such as HallusionBench and POPE do not include gaze annotations or surrogate attention signals, making it difficult to directly apply our method.
>
> That said, we recognize the importance of understanding whether our method reduces hallucination. To evaluate this, we conducted a human study comparing hallucination rates between the base and gaze-regularized models.
>
> We selected 200 examples from our dataset and generated outputs from the baseline model and the gaze-regularized model. These responses were randomly shuffled and presented to a human evaluator who was blind to which model produced which output. For each example, the evaluator was also shown the ground truth annotation, enabling a reliable comparison between the predicted output and the actual scene content. The evaluator was instructed to flag any instances where the model mentioned objects or actions that were either absent from the visual input or clearly unrelated.
> Taking inspiration from the metric in HallusionBench, we provide the $\mathcal{C}_I$ score which is the ratio of the annotations with clearly hallucinated instances divided by total number of annotations.
> | Model Variant    | Hallucinated Cases | Total Samples | $\mathcal{C}_I$ Score |
> | ---------------------- | ------------------ | ------------- | --------------------- |
> | Base model             | 41                | 200           | 0.205                 |
> | Gaze-regularized model | 28                | 200           | 0.140                |
>
> This shows that our proposed gaze regularized model produces an improvement by reducing the hallucination rate. We agree that extending our method to hallucination benchmarks is promising, and we will explore incorporating proxy gaze signals or human attention annotations for such benchmarks in future work.
>
> **(3) Related work on attention manipulation in VLMs:**
>
> We appreciate the suggestion to better contextualize our work with respect to recent attention-manipulation studies in multimodal models.
> Recent works such as OPERA and Perception in Reflection operate on the cross-attention maps during text generation and apply retrospective corrections if and when the model focuses on irrelevant regions. For instance, OPERA penalizes captions that mention objects like “pencil” if the cross-attention never highlighted any pencil in the image. Perception in Reflection similarly evaluates whether the model has missed out on finer details by reflecting on generated answers using a multi-turn dialogue dataset.
>
> In contrast, our work focuses on egocentric video understanding, where the goal is to model human behavior and intent in dynamic scenes. In this context, human gaze offers a strong and temporally grounded prior for what the person is doing or about to do as humans naturally fixate on task-relevant objects during interaction. Rather than relying on post-hoc attention correction, our method introduces gaze alignment to shape the visual representation passed into the language module, helping the model focus on semantically meaningful content from the start.
>
> Our approach is compatible with various transformer-based VLMs and does not require gaze at inference time. We will expand Section 2 or Appendix Sec. 6.1 to include these recent VLM attention regularization works.
>
> **(4) Which attention layers are regularized:**
>
> We apply gaze regularization to the final attention layer of a dedicated transformer block, which we insert after visual feature extraction (e.g., ViT) and before the fusion with the language decoder. This design ensures that gaze supervision directly shapes the final visual representation passed to the language module, guiding the model to focus on semantically relevant content. We chose this location to keep the gaze module architecturally modular and compatible across different VLMs. We also experimented with other placements and attention depths (Appendix Sec. 6.5.7), but found this setup offered the best trade-off between performance and flexibility.
>
> We thank the reviewer again for the helpful feedback and hope these clarifications strengthen your confidence in our submission. Please let us know if you have more questions.

---

> ### Comment · Area_Chair_utb6 · 2025-08-05
> **Please discuss rebuttal with authors**
>
> Please discuss rebuttal with authors

---

> ### Comment · Reviewer_qTVN · 2025-08-08
>
> Thank you for the thorough explanations and guidance. My concerns have been addressed, and I would like to maintain my initial support for this paper.

---

### Note · Authors · 2025-08-12

We thank the reviewers and the AC for the time, constructive feedback, and engagement during the discussion period. This process has helped us clarify our contributions and strengthen our submission.
Our core contribution is a modular gaze-regularized attention mechanism for egocentric VLMs, paired with an occlusion-aware temporal aggregation strategy to collect eye-gaze and incorporate human priors into VLMs. The approach uses gaze only during training, adding minimal inference overhead, and is compatible with a wide range of transformer-based VLMs. By aligning model attention with human visual focus via a KL-divergence loss, we guide the model toward task-relevant regions. We have addressed concerns on evaluation breadth by adding experiments on STA and EgoVQA using predicted gaze, demonstrating consistent gains beyond the two main gaze-annotated tasks. Only a subset of available egocentric datasets contain gaze annotations, making it difficult to directly apply our method to all benchmarks, but the added STA and EgoVQA experiments show the approach generalizes well even without real gaze. For theoretical justification, we clarified why temporally grounded gaze provides a useful prior for modeling human intent and spatial context, and showed that the top-10 attention–gaze overlap increases from 42% to 68%, indicating the model, in effect, learns to approximate gaze.
On related work, we expanded our discussion to include Min & Corso (WACV’21) and other gaze/attention-regularization studies, clarifying shared motivations as well as differences in scope, architecture, and inference requirements. We also detailed our human-in-the-loop GPT-4V annotation pipeline, which includes iterative prompt refinement and quality control of final outputs using human evaluation, alongside an analysis of partial gaze–caption alignment.

Additional results include a hallucination reduction from 20.5% to 14.0% and consistent performance improvements with both real and predicted gaze, reinforcing the method’s generality.

In summary, the method is simple, modular, and effective, offering a practical way to integrate human-like spatial priors into VLMs. The additional suggestions from the reviewers will be incorporated in the revised version, including expanded related work, the gaze–caption alignment study, and the top-k attention–gaze overlap analysis, to further strengthen clarity, transparency, and completeness of our work. We thank the reviewers again for the suggestions.

---

### Decision · Program_Chairs · 2025-09-17

**Decision:**

Accept (poster)

**Comment:**

The paper presents a gaze-regularized framework for VLMs on egocentric vision tasks. While one reviewer bTnR maintains a "borderline reject" (3) due to concerns about evaluation being limited to two tasks, the authors provided an explanation that the lack of gaze-annotated datasets inherently restricts broader task validation. The other reviewers found the rebuttal convincing, addressing concerns about metrics and model details. Therefore, my initial recommendation is acceptance.